# Large deviations in Coulomb gases: a mathematical perspective

**Mylène Maïda[1*], Antony Fahmy[2†], Jan-Luka Fatras[3‡], and Yilin Ye[4∘]**

**1** Univ. Lille, CNRS, Inria, UMR 8524 - Laboratoire Paul Painlevé, F-59000 Lille, France
**2** The Ohio State University, Department of Physics, Columbus OH, 43210
**3** Institut de Mathématiques, UMR5219, Université de Toulouse, CNRS, UPS,
F-31062 Toulouse Cedex 9, France
**4** Laboratoire de Physique de la Matière Condensée (UMR 7643),
CNRS – Ecole Polytechnique, Institut Polytechnique de Paris, 91120 Palaiseau, France

⋆ mylene.maida@univ-lille.fr ,    † fahmy.25@osu.edu ,    ‡ jan-luka.fatras@math.univ-toulouse.fr ,
∘ yilin.ye@polytechnique.edu

## Abstract

These notes account for five ninety-minute lectures given by Mylène Maïda as part of the 2024 Summer School in Les Houches. This 4-week program was entitled *Large deviations and applications*. The goal of these lectures is to present a series of mathematical results about large deviations of the particles of a Coulomb gas or related systems, such as the eigenvalues of some random matrix ensembles. It encompasses the deviations of the empirical measure and those of the rightmost particle (corresponding to the largest eigenvalue).

# Introduction

These notes account for five lectures given as part of the 2024 Summer School in Les Houches entitled *Large deviations and applications*. The goal of these lectures is to present a series of mathematical results that are known about large deviations of the empirical measure of the particles of a Coulomb gas or related systems, such as the eigenvalues of some random matrix ensembles.

The lectures were mostly taught in parallel with a course of the same format presented by Pierpaolo Vivo (King's College London) entitled *Large deviations in random matrix theory and Coulomb gas systems*, whose lecture notes can be found here[1]. We refer the interested reader to Vivo's notes for a complementary point of view on some of the results.

Among the five main courses of the program, this course was probably the most math-oriented. Therefore, along with the presentation of the results, we will also seize the opportunity to introduce some mathematical tools that we find useful to show (or use) large deviation principles (LDP).

Before presenting in more details the scope of these lectures, let us provide a few general references.

We start with two resources, that we find particularly accessible for beginners :

- as a first glimpse on large deviations, we recommend the following blogpost[2], which is the transcription of a tutorial taught by D. Chafaï at ICERM in 2018,

- in the same summer school, an introductory course on large deviations, with a special focus on statistical mechanics, was given by H. Touchette. We highly recommend his lecture notes [1].

Among probabilists, the following books are considered very classical:

- the book [2] provides a very comprehensive presentation of the main tools used to establish large deviation principles and of the most classical applications,

- the book [3] is a classical reference dealing with random matrix theory but we advertise here its appendix D as a very concise summary of useful tools for large deviations,

- the reference [4], which is also very comprehensive, is mainly based on a weak convergence approach, which, in its spirit, is more related to variational principles, that are natural to physicists and inspired the approach of D. García-Zelada, that we will present in Section 2.

These are general references for the course but more specific thematic lists of references will be provided in each chapter.

The structure of the present lecture notes is as follows : in Section 1 – corresponding to the first lecture – we will introduce one of the most studied ensembles of random matrices, the Gaussian Unitary Ensemble (GUE), provide an LDP for the empirical measure of its eigenvalues and explain how it can be exploited to recover the celebrated Wigner theorem in this particular case. This will mostly rely on a paper by G. Ben Arous and A. Guionnet [5]. In Section 2 – roughly corresponding to lectures 2 and 3 –, we advertise the work of D. García-Zelada [6], based on Varadhan's approach of large deviations, that provides a unified framework for large

---

[1]http://www.lptms.universite-paris-saclay.fr/leshouches2024/files/2024/07/Les_Houches_Lecture_Notes_VIVO_V1.pdf

[2]https://djalil.chafai.net/blog/2018/03/09/tutorial-on-large-deviation-principles/

deviations for singular Gibbs measures, encompassing usual Coulomb gases in $\mathbb{R}^d$ at finite or high temperature, but also Coulomb gases on manifolds, conditional Gibbs measures, zeroes of some models of random polynomials etc. Recently, following the pioneering work of Guionnet and Husson, spherical integrals of the form

$$I_N(A_N, B_N) := \int \exp\left(N\mathrm{Tr}(A_N U B_N U^*)\right) \mathrm{d}m_N(U),  \tag{1}$$

where $A_N$ and $B_N$ are two diagonal matrices of size $N$ with real entries and $m_N$ is the Haar measure on the orthogonal or the unitary group of size $N$, have been used to study the large deviations of the largest eigenvalue for several models of random matrices. In Section 3 – roughly corresponding to lectures 4 and 5 –, we provide a detailed derivation of the asymptotics of spherical integrals in the case when one of the matrices, say $A_N$, is of rank one, and explain how it can be used to study the deviations of the largest eigenvalue.

In these notes, we try to stay as close as possible to the in-person lectures that have been given in Les Houches. For the sake of completeness, we have nevertheless added a few proofs that were not presented during the lectures: they are in general postponed to the appendices.

Note that, although very interesting, the results on large deviations of the empirical field for Coulomb gases [7,8], which are related to the microscopic structure of these particle systems, are beyond the scope of this course and will not be included in these notes.

# 1  The Gaussian Unitary Ensemble

The *Gaussian Unitary Ensemble (GUE)* is one of the most popular models of random matrices. In this first chapter, we study this example in full detail, through the lens of large deviation theory.

## 1.1  Three descriptions of the GUE

In the usual vocabulary of random matrix theory (RMT), inspired by statistical physics, an *ensemble* is a probability distribution over a set of matrices. In this case, we consider the space of Hermitian matrices of size $N \times N$, denoted by

$$\mathcal{H}_N(\mathbb{C}) := \{M \in \mathcal{M}_N(\mathbb{C}), M^* = M\}.$$

The easiest way to define the GUE is by describing the joint law of the entries. Before doing so, we recall that if $X$ and $Y$ are two independent real random variables with standard Gaussian distribution $\mathcal{N}(0,1)$, then $G := \frac{X+\mathrm{i}Y}{\sqrt{2}}$ is said to be *standard complex Gaussian* and we denote $G \sim \mathcal{N}_\mathbb{C}(0,1)$.

**Definition 1.1** *Let $N \in \mathbb{N}^*$ and consider independent random variables $\{G_{i,i}\}_{i=1}^N$ and $\{G_{i,j}\}_{1 \le i < j \le N}$ such that $G_{i,i} \sim \mathcal{N}(0,1)$ and $G_{i,j} \sim \mathcal{N}_\mathbb{C}(0,1)$. Define the following $N \times N$ Hermitian matrix :*

$$H_N = \begin{pmatrix} \frac{G_{1,1}}{\sqrt{N}} & & \frac{G_{i,j}}{\sqrt{N}} & \\ & \ddots & & \\ \frac{G_{i,j}^*}{\sqrt{N}} & & \ddots & \\ & & & \frac{G_{N,N}}{\sqrt{N}} \end{pmatrix}, \quad so\ that\ G_{i,j} = G_{j,i}^*.$$

*The matrix $H_N$ is said to follow the GUE distribution or equivalently to belong to the GUE. We denote by $\mathbb{P}_{GUE_N}$ its distribution.*

One can also directly define $\mathbb{P}_{GUE_N}$ as a Gaussian distribution on $\mathcal{H}_N(\mathbb{C})$. The isomorphism $\mathcal{H}_N(\mathbb{C}) \simeq \mathbb{R}^{N^2}$ induces a Lebesgue measure on $\mathcal{H}_N(\mathbb{C})$, that we denote by $\mathbf{Leb}_{\mathcal{H}_N}$. We can then give the following equivalent definition of the GUE:

**Proposition 1.2** *There exists a normalizing constant $c_N$ such that*

$$\mathrm{d}\mathbb{P}_{GUE_N}(H) = c_N \exp\left(-\frac{N}{2}Tr(H^2)\right) \mathrm{d}\mathbf{Leb}_{\mathcal{H}_N}(H),$$

*where $Tr$ is the usual trace on $\mathcal{H}_N(\mathbb{C})$.*

To see the correspondence with the law of the entries, it is enough to expand the trace as follows: if $H = (h_{i,j})_{1 \le i,j \le N}$,

$$\mathrm{Tr}(H^2) = \mathrm{Tr}(HH^*) = \sum_{i=1}^N h_{i,i}^2 + 2 \sum_{1 \le i < j \le N} |h_{i,j}|^2.$$

Now, for $i < j$, if we denote by $x_{i,j} = \mathbf{Re}\,h_{i,j}$ and $y_{i,j} = \mathbf{Im}\,h_{i,j}$, the respective real and imaginary part of $h_{i,j}$, we have

$$\mathrm{Tr}(H^2) = \sum_{i=1}^N h_{i,i}^2 + 2 \sum_{1 \le i < j \le N} (x_{i,j}^2 + y_{i,j}^2),$$

so that, as expected, under $\mathbb{P}_{GUE_N}$, $(h_{i,i})_{1 \leq i \leq N}$, $(x_{i,j})_{1 \leq i < j \leq N}$ and $(y_{i,j})_{1 \leq i < j \leq N}$ are independent real Gaussian variables, with variance $1/N$ if $i = j$ and $1/2N$ if $i < j$.

When $H_N$ has distribution $\mathbb{P}_{GUE_N}$, it is interesting to study the law of its eigenvalues and eigenvectors. The following proposition gives the distribution of the eigenvalues. By a slight abuse of notations[3], we will again denote this joint distribution by $\mathbb{P}_{GUE_N}$.

**Proposition 1.3** *If $H_N$ has distribution $\mathbb{P}_{GUE_N}$, then almost surely, $H_N$ is diagonalisable with distinct eigenvalues, that we may enumerate in decreasing order $\lambda_1^N > \cdots > \lambda_N^N$. Then, the joint law of the random vector $(\lambda_1^N, \cdots, \lambda_N^N)$ is given by*

$$d\mathbb{P}_{GUE_N}(\lambda_1, \cdots, \lambda_N) = \frac{N^{\frac{N^2}{2}} \mathbf{1}_{\{\lambda_1 > \ldots > \lambda_N\}}}{(2\pi)^{N/2} \prod_{j=1}^{N-1} j!} \prod_{i<j} (\lambda_i - \lambda_j)^2 \exp\left(-\frac{N}{2} \sum_{j=1}^{N} \lambda_j^2\right) d\lambda_1 \cdots d\lambda_N. \quad (2)$$

This statement is well known in RMT. It is closely related to Weyl's formula. A classical reference for this kind of results is the book of M.L. Mehta [9]. One can also cite [10] for a gentle introduction for physicists. For probabilists, a more recent standard reference is [3] (see in particular Theorem 2.5.2 there).

Although we won't focus very much on this aspect in the sequel, let us mention that it is also possible to describe the law of the eigenvectors under $\mathbb{P}_{GUE_N}$. The answer to this question, together with a third description of the law $\mathbb{P}_{GUE_N}$, is postponed to Appendix A.

We now want to study the behavior of the particles $(\lambda_1^N, \cdots, \lambda_N^N)$ under $\mathbb{P}_{GUE_N}$. Many interesting questions can be asked about their behavior e.g. the following:

- How does the largest eigenvalue behave ?

- What does the global regime look like ? etc.

The first question will be addressed in full detail for the Gaussian Orthogonal Ensemble (GOE), which is the real symmetric counterpart of the GUE, in the course of Pierpaolo Vivo and we strongly recommend his lecture notes. They can be found in the present volume or at the following link[4]. We won't detail it in the case of the GUE, but we will come back to similar questions for other models in the third section of these notes (Lectures 4 and 5).

We will rather focus on the second question. The idea is to encode the positions of all the particles as a whole in the following object:

$$\hat{\mu}_N := \frac{1}{N} \sum_{i=1}^{N} \delta_{\lambda_i^N}.$$

It is called the *empirical distribution of the eigenvalues of $H_N$* or *spectral empirical distribution of $H_N$*. For each realisation $H_N(\omega)$ of the random matrix $H_N$, $\hat{\mu}_N(\omega)$ is a probability measure which is nothing but the uniform distribution over the set of eigenvalues $\{\lambda_1^N(\omega), \cdots, \lambda_N^N(\omega)\}$. Therefore, $\hat{\mu}_N$ is a *random probability measure*, that is a random variable with values in the set $\mathcal{P}(\mathbb{R})$ of probability measures on $\mathbb{R}$. This random measure will be our main object of study in this chapter, and we want in particular to describe its typical behavior (law of large numbers), its large deviations etc.

---

[3]If $H$ is a matrix, $d\mathbb{P}_{GUE_N}(H)$ will refer to the law of the matrix, whereas when $\lambda_1, \lambda_N$ are real numbers, $d\mathbb{P}_{GUE_N}(\lambda_1, \cdots, \lambda_N)$, it will refer to the law of the eigenvalues, so that there is hopefully no ambiguity.

[4]http://www.lptms.universite-paris-saclay.fr/leshouches2024/files/2024/07/Les_Houches_Lecture_Notes_VIVO_V1.pdf

## 1.2   Large deviation principle for the empirical spectral distribution

Let us first make the link between the GUE random matrix model and Coulomb gas-like particle systems.

To lighten the notations, we denote the prefactor in (2) by

$$C_N := \frac{N^{\frac{N^2}{2}}}{(2\pi)^{N/2} \prod_{j=1}^{N} j!},$$

(3)

so that we can now rewrite (2) as :

$$d\mathbb{P}_{GUE_N}(\lambda_1, \ldots, \lambda_N) = C_N \exp\left(-N\left(\frac{1}{2}\sum_{j=1}^{N}\lambda_j^2 - \frac{1}{N}\sum_{i\neq j}\log|\lambda_i - \lambda_j|\right)\right)d\lambda_1 \cdots d\lambda_N.$$

Note that there is here a slight abuse of notation: $\mathbb{P}_{GUE_N}$ as defined in (2) was a distribution over that set $\{\lambda_1 > \ldots > \lambda_N\}$ whereas here we extend it to $\mathbb{R}^N$. This is balanced by an extra factor $N!$ in the definition of the constant $C_N$ with respect to the normalizing constant appearing in (2).

We now can see $\mathbb{P}_{GUE_N}$ as the *canonical Gibbs measure* associated to the energy $E$, defined as follows: for any $N$-tuple $x_1, \ldots, x_N$ of real numbers,

$$E(x_1, \ldots, x_N) := N\left(\frac{1}{2}\sum_{j=1}^{N}x_j^2 - \frac{1}{N}\sum_{i\neq j}\log|x_i - x_j|\right).$$

(4)

In this expression,

- the first term $\frac{1}{2}\sum_{j=1}^{N}x_j^2$ is usually interpreted as a confining external potential applied to each particle, that prevents them to lay too far away from the origin,

- whereas the second term $\frac{1}{N}\sum_{i\neq j}\log|x_i - x_j|$ is usually interpreted as a repulsive two-body interaction.

We commonly use the terminology *one dimensional log-gas* to describe such a particle system; it is considered a Coulomb-type particle system[5]. Coulomb gases will be introduced and discussed more thoroughly in the next chapter of these notes. We refer to the book [11] of P. Forrester for a very thorough presentation of theses systems, including many explicit computations.

Before getting into the mathematical statement of an LDP for the spectral empirical measure $\widehat{\mu}_N$, let us try to give some rough heuristics towards a possible rate function. Fix $\mu \in \mathcal{P}(\mathbb{R})$, $\delta > 0$ small and $B(\mu, \delta)$ a ball of radius $\delta$ centered at $\mu$ for a metric on $\mathcal{P}(\mathbb{R})$ to be defined later. We have

$$\mathbb{P}_{GUE_N}(\widehat{\mu}_N \in B(\mu, \delta))$$

$$= C_N \int_{\widehat{\mu}_N \in B(\mu, \delta)} \exp\left(-N^2\left(\int \frac{x^2}{2}d\widehat{\mu}_N(x) - \iint_{x\neq y}\log|x-y|d\widehat{\mu}_N(x)d\widehat{\mu}_N(y)\right)\right)dx_1 \cdots dx_N.$$

Then (if everything behaves nicely)

$$-\frac{1}{N^2}\log\mathbb{P}_{GUE}(\widehat{\mu}_N \in B(\mu, \delta)) \approx -\frac{1}{N^2}\log C_N + \int \frac{x^2}{2}d\mu(x) - \iint \log|x-y|d\mu(x)d\mu(y).$$

---

[5]The one-dimensional Coulomb interaction is linear whereas the two-dimensional is logarithmic. In other words, we have here a two-dimensional Coulomb gas confined to live on the real line.

The analysis of the constant $C_N$ is a simple exercise, as its expression is completely explicit. Namely,

$$\frac{1}{N^2}\log C_N = -\frac{1}{N^2}\sum_{j=1}^{N}\sum_{k=1}^{j}\log\left(\frac{k}{N}\right) - \frac{1}{2N}\log\left(\frac{2\pi}{N}\right)$$

$$= -\frac{1}{N}\sum_{k=1}^{N}\frac{N-k+1}{N}\log\left(\frac{k}{N}\right) - \frac{1}{2N}\log\left(\frac{2\pi}{N}\right)$$

$$\xrightarrow[N\to\infty]{} -\int_{0}^{1}(1-x)\log x\,\mathrm{d}x = \frac{3}{4}.$$

If, for any probability measure $\mu$ for which it is properly defined, we let

$$I(\mu) = \int \frac{x^2}{2}\mathrm{d}\mu(x) - \iint \log|x-y|\mathrm{d}\mu(x)\mathrm{d}\mu(y) - \frac{3}{4},$$

then we expect that

$$\mathbb{P}_{GUE_N}(\widehat{\mu}_N \in B(\mu,\delta)) \simeq \exp\left(-N^2 I(\mu)\right).$$

Let us now go to a more precise statement of the LDP that was unveiled by G. Ben Arous and A. Guionnet in [5]. Mathematically speaking, a full LDP in this case will take the following form:

- for any open set $O \subset \mathcal{P}(\mathbb{R})$, $\liminf_{N\to\infty}\frac{1}{N^2}\log\mathbb{P}_{\mathrm{GUE}_N}(\widehat{\mu}_N \in O) \geq -\inf_{\mu\in O} I(\mu)$,

- for any closed set $F \subset \mathcal{P}(\mathbb{R})$, $\limsup_{N\to\infty}\frac{1}{N^2}\log\mathbb{P}_{\mathrm{GUE}_N}(\widehat{\mu}_N \in F) \leq -\inf_{\mu\in F} I(\mu)$.

*Open* and *closed* refer to a topology that we have to define on the space of probability measures $\mathcal{P}(\mathbb{R})$: a common choice is the topology of weak convergence. In this topology, a sequence $(\nu_N)_{N\in\mathbb{N}}$ converges to $\nu \in \mathcal{P}(\mathbb{R})$, and we denote this convergence by $\nu_N \xrightarrow[N\to\infty]{w} \nu$, if and only if

$$\forall f \in \mathcal{C}^b(\mathbb{R}), \int_{\mathbb{R}} f(x)\mathrm{d}\nu_N(x) \xrightarrow[N\to\infty]{} \int_{\mathbb{R}} f(x)\mathrm{d}\nu(x),$$

where $\mathcal{C}^b(\mathbb{R})$ stands for the set of bounded and continuous functions from $\mathbb{R}$ to $\mathbb{R}$.

We are now ready to state the main result of this chapter.

**Theorem 1.4** *[5] Under $\mathbb{P}_{GUE_N}$, the sequence of empirical spectral distributions $(\widehat{\mu}_N)_{N\in\mathbb{N}}$ satisfies a large deviation principle at speed $N^2$ with good rate function[6] $I$ in the space $\mathcal{P}(\mathbb{R})$ equipped with the topology of weak convergence, where the rate function $I$ is defined as follows:*

$$I(\mu) := \begin{cases} \int_{\mathbb{R}} \frac{x^2}{2}\mathrm{d}\mu(x) - \iint \log|x-y|\mathrm{d}\mu(x)\mathrm{d}\mu(y) - \frac{3}{4}, & \text{if } \int x^2\mathrm{d}\mu < \infty, \\ \infty, & \text{otherwise.} \end{cases} \tag{5}$$

It is always more comfortable to work with a metric structure. Fortunately, the topology of weak convergence can be metrized by the bounded-Lipschitz distance defined as follows : for $\mu,\nu \in \mathcal{P}(\mathbb{R})$

$$d_{\mathrm{BL}}(\mu,\nu) = \sup_{\|f\|_\infty \leq 1, \|f\|_{\mathrm{Lip}} \leq 1}\left|\int f\mathrm{d}\mu - \int f\mathrm{d}\nu\right|,$$

---

[6] We don't want to insist too much at this stage on the notion of *(good) rate function*, we refer to Section 2.3 for more details.

with $\|f\|_{\text{Lip}} \leq 1 \Leftrightarrow |f(x) - f(y)| \leq |x - y|, \forall x, y \in \mathbb{R}$. This means that $\nu_N \xrightarrow[N \to \infty]{w} \nu$ if and only if $d_{\text{BL}}(\nu_N, \nu) \xrightarrow[N \to \infty]{} 0$. In the following, anytime we mention a distance on $\mathcal{P}(\mathbb{R})$ it will be the bounded-Lipschitz distance and $B(\mu, \delta)$ will refer to the ball of radius $\delta$ around $\mu$ for this bounded-Lipschitz distance.

With $\mathcal{P}(\mathbb{R})$ being a metric space, it is possible to give an easier formulation of the LDP above. Roughly speaking, we have :

(weak LDP on small balls + exponential tightness) implies (full LDP)

More precisely, if we have

1. (Weak LDP) :

$$\lim_{\delta \to 0} \liminf_{N \to \infty} \frac{1}{N^2} \log \mathbb{P}_{\text{GUE}_N}(\widehat{\mu}_N \in B(\mu, \delta))$$
$$= \lim_{\delta \to 0} \limsup_{N \to \infty} \frac{1}{N^2} \log \mathbb{P}_{\text{GUE}_N}(\widehat{\mu}_N \in B(\mu, \delta)) =: -I(\mu),$$

2. and (Exponential tightness) : There exists a sequence $(K_L)_{L>0}$ of compact subsets of $\mathcal{P}(\mathbb{R})$ such that

$$\limsup_{L \to \infty} \limsup_{N \to \infty} \frac{1}{N^2} \log \mathbb{P}_{\text{GUE}_N}(\widehat{\mu}_N \notin K_L) = -\infty, \tag{6}$$

then we have Theorem 1.4. We refer to Appendix D in [3] for the details on this criterion.

The proof of the weak LDP has been sketched at the beginning of this subsection, so we now focus on the second point. Let us comment on the (important) notion of *tightness* in probability. A classical reference on the notions of weak convergence of measures and tightness in Polish spaces in the book of P. Billingsley [12]. If we have a random variable $X$, with values in a Polish space, then for any $\varepsilon > 0$, one can always find a compact set $\mathcal{K}_\varepsilon$ for which

$$\mathbb{P}(X \notin \mathcal{K}_\varepsilon) \leq \varepsilon,$$

that is "almost everything happens inside a (large enough) compact set". When we consider a sequence, or more generally a family, of random variables $(X_i)_{i \in I}$, it is not obvious that one can find a fixed compact $\mathcal{K}_\varepsilon$ (depending on $\varepsilon$ but not on $i$) such that

$$\forall i \in I, \mathbb{P}(X_i \notin \mathcal{K}_\varepsilon) \leq \varepsilon.$$

This is not true in general[7]. If it holds for any $\varepsilon$, the family of random variables is said to be *tight* (equivalently, if for any $i \in I$, $\mu_i$ is the distribution of the random variable $X_i$, the family of probability measures $(\mu_i)_{i \in I}$ is said to be tight). It means that when we deal with questions related to (weak) convergence, what happens outside a large compact set is not relevant.

Here, as we are working at the level of exponentially small events, we ask for *exponential tightness*, which, in our case, is expressed by (6). Moreover, as the sequence $(\widehat{\mu}_N)_{N \geq 1}$ that we are considering is a sequence of random variables with values in the set $\mathcal{P}(\mathbb{R})$, the first step is to describe a convenient family of compact sets in this latter space.

---

[7]A simple illustrative example is the case when the distribution of $X_n$ is $\mu_n = \delta_n$ the Dirac mass at $n$. This sequence of probability measures converges to the null measure in the topology of vague convergence (for test functions which are continuous and compactly supported) but does not converge in the sense of weak convergence. We observe a loss of mass due to the lack of tightness.

227    For any $L > 0$, let us define

$$K_L := \left\{ \mu \in \mathcal{P}(\mathbb{R}), \int x^2 \mathrm{d}\mu(x) \le L \right\}.$$

228   We first justify that $K_L$ is a compact subset of $\mathcal{P}(\mathbb{R})$. Notice that for all $\mu \in K_L$, we have by
229   Markov inequality

$$\mu\left( \left[ -\sqrt{\frac{L}{\varepsilon}}, \sqrt{\frac{L}{\varepsilon}} \right]^c \right) = \frac{\varepsilon}{L} \int \frac{L}{\varepsilon} \mathbb{1}_{\left\{ x \notin \left[ -\sqrt{\frac{L}{\varepsilon}}, \sqrt{\frac{L}{\varepsilon}} \right] \right\}}(x) \mathrm{d}\mu(x) \le \frac{\varepsilon}{L} \int x^2 \mathrm{d}\mu(x) \le \varepsilon,$$

230   so that the family of probability measure $K_L$ is tight, in the sense explained above[8]. Since
231   $\mathbb{R}$ is a complete metric space, we deduce by Prokhorov's theorem (see for example Theorem
232   C.9 in [3]) that the closure of $K_L$ is compact in the weak topology. Moreover, $K_L$ is closed.
233   Indeed, let $(\mu_N)_N$ be a sequence in $K_L$ which converges weakly to $\mu$ then, for any $M > 0$,
234   $\int \min(x^2, M) \mathrm{d}\mu(x) = \lim_{N \to \infty} \int \min(x^2, M) \mathrm{d}\mu_N(x)$. Then by monotone convergence as $M$
235   goes to infinity and the fact that the bound $\int \min(x^2, M) \mathrm{d}\mu_N(x) \le L$ is uniform in $M$ and
236   $N$, we get that $\mu \in K_L$. Therefore, $K_L$ is a closed set included in a compact and so it is itself
237   compact.
238    Let us now show (6). We define

$$f(x, y) := \frac{x^2}{4} + \frac{y^2}{4} - \log|x - y|.$$

239   As, for any $x, y \in \mathbb{R}$, $\log|x - y| \le \log(|x| + 1) + \log(|y| + 1)$, we have

$$f(x, y) \ge \frac{x^2}{8} + \frac{y^2}{8} + \widetilde{C},$$

240   for some constant $\widetilde{C}$. Note that this bound also justifies why the rate function $I$ introduced in
241   (5) is well defined.
242    Moreover, using the density of $\mathbb{P}_{GUE_N}$ with respect to the Lebesgue measure, we have :

$$\mathbb{P}_{\mathrm{GUE}_N}(\widehat{\mu}_N \notin K_L) = C_N \int_{\{\widehat{\mu}_N \notin K_L\}} \exp\left( -N \frac{1}{2} \sum_{i=1}^{N} x_i^2 + \sum_{i \ne j} \log|x_i - x_j| \right) \mathrm{d}x_1 \ldots \mathrm{d}x_N$$

$$= C_N \int_{\{\widehat{\mu}_N \notin K_L\}} \exp\left( -N^2 \iint_{x \ne y} f(x, y) \mathrm{d}\widehat{\mu}_N(x) \mathrm{d}\widehat{\mu}_N(y) \right) \prod_{i=1}^{N} \exp\left( -\frac{x_i^2}{2} \right) \mathrm{d}x_1 \cdots \mathrm{d}x_N$$

$$\le C_N \int_{\{\widehat{\mu}_N \notin K_L\}} \exp\left( -N^2 \iint_{x \ne y} \left( \frac{x^2}{8} + \frac{y^2}{8} + \widetilde{C} \right) \mathrm{d}\widehat{\mu}_N(x) \mathrm{d}\widehat{\mu}_N(y) \right) \prod_{i=1}^{N} \exp\left( -\frac{x_i^2}{2} \right) \mathrm{d}x_1 \cdots \mathrm{d}x_N$$

$$\le C_N \exp\left( -N^2 \left( \frac{N-1}{N} \frac{L}{4} + \frac{N(N-1)}{N^2} \widetilde{C} \right) \right) \int_{\{\widehat{\mu}_N \notin K_L\}} \prod_{i=1}^{N} \exp\left( -\frac{x_i^2}{2} \right) \mathrm{d}x_1 \cdots \mathrm{d}x_N,$$

243   and then taking $\limsup \frac{1}{N^2} \log$ on both sides, we get :

$$\limsup_{N \to \infty} \frac{1}{N^2} \log \mathbb{P}_{\mathrm{GUE}_N}(\widehat{\mu}_N \notin K_L) \le \limsup_{N \to \infty} \frac{1}{N^2} \log C_N - \frac{L}{4} + \widetilde{C} \le \frac{3}{4} - \frac{L}{4} + \widetilde{C}.$$

---

[8]Note that there is a subtle point here: we use the fact that the family of probability measure $K_L$ is tight to show that it is a compact subset of $\mathcal{P}(\mathbb{R})$. Then we will use $K_L$ to show that the family of random variables $(\widehat{\mu}_N)_{N \ge 1}$ is exponentially tight !

244    Finally, taking $L$ to infinity, we get the desired result.

245

246    This concludes the arguments of the proof of the result of G. Ben Arous and A. Guionnet
247 that we wanted to emphasize here. We refer to the original paper [5] or alternatively to Section
248 2.6 of the book [3] for more details. In the framework of these notes, we will give in the next
249 chapter a much more general result on singular Gibbs measures that encompasses the GUE
250 model.

## 1.3   Understanding the minimizer of the rate function

252 In various situations, understanding the deviations of a family of random variables may be
253 the best way to study also their typical behavior. In the case of GUE, this typical behavior
254 was known for a long time before the large deviations were studied but we find it instructive
255 to show how this particular case of Wigner's theorem can be seen as a corollary of the large
256 deviation principle we have just obtained. This subsection will be devoted to the discussion
257 and the proof of the following statement and why it may be seen as a corollary of Theorem
258 1.4.

259 **Corollary 1.5** *(Wigner's Theorem)*
260   *Almost surely*

$$\widehat{\mu}_N \xrightarrow[N \to \infty]{w} \mu_{sc},$$

261 *where $\mu_{sc}$ is the semi-circular distribution defined by the density :*

$$\mathrm{d}\mu_{sc}(t) = \frac{1}{2\pi} \sqrt{4 - t^2} \mathbf{1}_{[-2,2]}(t) \mathrm{d}t.$$

262    In a very general context, it is possible to deduce an almost sure convergence from a large
263 deviation principle, whenever the rate has a unique minimizer. This general mechanism will be
264 illustrated in our example at the end of this section. We first establish the following property:

265 **Proposition 1.6** $\mu_{sc}$ *is the unique minimizer of $I$, the rate function defined in* (5).

266    The proof of the proposition will be in three steps: we show that any minimizer should
267 satisfy the Euler-Lagrange equations, that the semi-circular distribution satisfies the Euler-
268 Lagrange equations and to conclude, that the minimizer is unique.
269    Each of the three steps corresponds to a lemma that we state below:

270 **Lemma 1.7** *Any minimizer $\mu$ of the rate function $I$ defined in* (5) *satisfies the following : there*
271 *exists a constant $C_{EL}$ such that for any $x$ in the support of the measure $\mu$, we have*

$$\frac{x^2}{2} - 2 \int \log|x - y| \mathrm{d}\mu(y) = C_{EL},$$

272 *and for Lebesgue-almost every $x \in \mathbb{R}$,*

$$\frac{x^2}{2} - 2 \int \log|x - y| \mathrm{d}\mu(y) \geq C_{EL}.$$

273    These equations are called *Euler-Lagrange (EL) equations*. We will give below a detailed
274 proof of Lemma 1.7, which, as we will see, is robust to generalisation to external potentials
275 other than quadratic.
276    The next lemma states that $\mu_{sc}$ does satisfy the EL-equation associated to this problem :

**Lemma 1.8**

$$\frac{x^2}{2} - 2\int \log|x-y| \mathrm{d}\mu_{\mathrm{sc}}(y) = \begin{cases} 1 & \text{for all } x \in [-2,2], \\ > 1 & \text{for all } |x| > 2. \end{cases}$$

There are many ways to compute the logarithmic potential of $\mu_{\mathrm{sc}}$, that is the integral $\int \log|x-y| \mathrm{d}\mu_{\mathrm{sc}}(y)$. The computation of this quantity outside the support of $\mu_{\mathrm{sc}}$ has been detailed in Section IV.A.1 of Vivo's lecture notes[9]: using an expansion of the logarithm, the computation boils down to the computation of the moments of $\mu_{\mathrm{sc}}$, that are interesting quantities by themselves, related to Catalan numbers. From his computation, it is easy to check the second inequality above. For the sake of completeness, we present the details of the computation of the logarithmic potential inside the support of the measure, using the residue theorem, in Appendix B.

Moreover, the uniqueness of the minimizer of the rate function is ensured by the following:

**Lemma 1.9** *The rate function $I$ defined in* (5) *is strictly convex on $\mathcal{P}(\mathbb{R})$. It therefore admits a unique minimizer.*

This was not proved during the lectures but relies on an interesting Fourier representation of the logarithmic energy : the proof of Lemma 1.9 is postponed to Appendix C.

We now go to the proof of Lemma 1.7. Let $\psi \geq 0$, and $\phi$ be two bounded and compactly supported functions. Then define $\overline{\nu}_{\psi,\phi}$ by

$$\mathrm{d}\overline{\nu}_{\psi,\phi}(x) = \phi(x)\mathrm{d}\mu(x) + \psi(x)\mathrm{d}x,$$

where $\phi$ and $\psi$ are such that $\overline{\nu}_{\psi,\phi}(\mathbb{R}) = 0$, so that if $\mu \in \mathcal{P}(\mathbb{R})$ and $\varepsilon$ is sufficiently small, $\mu + \varepsilon\overline{\nu}_{\psi,\phi} \in \mathcal{P}(\mathbb{R})$ . If $\mu$ is a minimizer of $I$, for any such $\psi, \phi$ we have

$$\begin{aligned} I(\mu) \leq I(\mu + \epsilon\overline{\nu}_{\psi,\phi}) &= \int \frac{x^2}{2}\mathrm{d}\mu + \epsilon\int \frac{x^2}{2}\mathrm{d}\overline{\nu}_{\psi,\phi} \\ &\quad - \iint \log|x-y|(\mathrm{d}\mu\mathrm{d}\mu + \epsilon\mathrm{d}\mu\mathrm{d}\overline{\nu}_{\psi,\phi} + \epsilon\mathrm{d}\overline{\nu}_{\psi,\phi}\mathrm{d}\mu + \epsilon^2\mathrm{d}\overline{\nu}_{\psi,\phi}\mathrm{d}\overline{\nu}_{\psi,\phi}) \\ &\quad - \frac{3}{4}, \end{aligned}$$

thus we get

$$\epsilon\int \frac{x^2}{2}\mathrm{d}\overline{\nu}_{\psi,\phi}(x) - 2\epsilon\iint \log|x-y|\mathrm{d}\overline{\nu}_{\psi,\phi}(x)\mathrm{d}\mu(y) - \epsilon^2\iint \log|x-y|\mathrm{d}\overline{\nu}_{\psi,\phi}(x)\mathrm{d}\overline{\nu}_{\psi,\phi}(y) \geq 0,$$

and so by dividing by $\varepsilon$ and letting $\varepsilon$ go to zero we get :

$$\int \left(\frac{x^2}{2} - 2\int \log|x-y|\mathrm{d}\mu(y)\right)\mathrm{d}\overline{\nu}_{\psi,\phi}(x) \geq 0.$$

By choosing $\psi = 0$ and $\pm\phi$, we obtain that for all $\phi$ such that $\int \phi\mathrm{d}\mu = 0$ :

$$\int \phi(x)\left(\frac{x^2}{2} - 2\int \log|x-y|\mathrm{d}\mu(y)\right)\mathrm{d}\mu(x) = 0,$$

---

[9]http://www.lptms.universite-paris-saclay.fr/leshouches2024/files/2024/07/Les_Houches_Lecture_Notes_VIVO_V1.pdf

299 and therefore there exists a constant $C_{EL}$ such that for all $x \in \text{Supp}(\mu)$,

$$\frac{x^2}{2} - 2 \int \log|x - y| \mathrm{d}\mu(y) = C_{EL} \text{ (Lagrange multiplier)}.$$

300 Then, by choosing $\phi = -\int \psi(y)\mathrm{d}y$ being a constant, we get, for Lebesgue-almost every $x$,

$$\frac{x^2}{2} - 2 \int \log|x - y| \mathrm{d}\mu(y) \geq C_{EL}.$$

301 Therefore, any minimizer $\mu$ satisfies the Euler-Lagrange equation.

302

303 We are now ready to go to the proof of Corollary 1.5. Putting the three lemmas together,
304 we get that $\mu_{\mathrm{sc}}$ is indeed the unique minimizer of $I$.

305

306 From there, one can easily deduce Wigner's theorem, using first the upper bound of the
307 large deviation principle. It indeed gives that

$$\forall \delta > 0, \limsup_{N \to \infty} \frac{1}{N^2} \log \mathbb{P}_{GUE_N}(\widehat{\mu}_N \notin B(\mu_{\mathrm{sc}}, \delta)) \leq - \inf_{\mu \notin B(\mu_{\mathrm{sc}}, \delta)} I(\mu) =: -I_\delta.$$

308 Then since $K := B(\mu_{sc}, \delta)^c \cap \{\nu : I(\nu) \leq I_\delta + 1\}$ (where $B(\mu_{sc}, \delta)^c$ is the complement of
309 $B(\mu_{sc}, \delta)$) is a compact set and $I$ is lower semicontinous, $I$ reaches its infimum on $K$. Since
310 $K$ does not contain the minimizer $\mu_{\mathrm{sc}}$ of $I$, we have $0 < \inf_{\mu \in K} I(\mu) = I_\delta$. Therefore, we have
311 for $N$ big enough that

$$\mathbb{P}_{GUE_N}(\widehat{\mu}_N \notin B(\mu_{sc}, \delta)) \leq \exp\left(-N^2 \frac{I_\delta}{2}\right),$$

312 and thus, since $I_\delta > 0$, we have that $\mathbb{P}_{GUE_N}(\widehat{\mu}_N \notin B(\mu_{\mathrm{sc}}, \delta))$ is summable. By Borel-Cantelli,
313 we know that for all $\delta$, the sequence $(\widehat{\mu}_N)_{N \in \mathbb{N}}$ is almost surely eventually in $B(\mu_{\mathrm{sc}}, \delta)$ and
314 therefore we have that a.s. $\widehat{\mu}_N \xrightarrow{w} \mu_{\mathrm{sc}}$ as $N \to \infty$.

## 1.4 Conclusion

316 Before going to the general theory of the global behavior of Coulomb gases, let us summarize
317 what we have learnt from the study of the specific case of the GUE model:

318 • If $H_N$ is a random matrix from the GUE of size $N$, the distribution of its eigenvalues is a
319   singular canonical Gibbs measure which forms a one-dimensional log-gas.

320 • Its spectral empirical distribution is a random measure which satisfies a large deviation
321   principle on the space of probability measures on $\mathbb{R}$, at speed $N^2$ with an explicit rate
322   function.

323 • Through the derivation of Euler-Lagrange equations, one can show that the unique min-
324   imizer of this rate function is the semi-circular distribution. From there, one can use the
325   large deviation upper bound for the spectral empirical distribution to get the almost sure
326   weak convergence of the latter to the semi-circle distribution (Wigner's theorem).

## 2  General LDP for particle systems related to Coulomb gases

After this warmup through the example of the GUE, we now go to the main topic of the course, that is LDPs for Coulomb gases and related particle systems. On this question, it is fair to cite the work of D. Chafaï, N. Gozlan and P. A. Zitt [13], which built on arguments in the spirit of [5]. We have chosen in this course to emphasize the work of D. García-Zelada [6]. We first introduce properly the notion of Coulomb gas.

### 2.1  Coulomb and Riesz gases, vocabulary

Consider $N$ particles $(x_1, \cdots, x_N) \in (\mathbb{R}^d)^N$, and define the Hamiltonian of the configuration as follows:

$$E_N(x_1, \cdots, x_N) = N \sum_{i=1}^{N} V(x_i) + \frac{1}{2} \sum_{i \neq j} g(x_i - x_j). \tag{7}$$

The function $V$ is usually called the *external potential* and $g$ the *kernel interaction*. Under appropriate assumptions on $V$ and $g$ that we will detail later, it is possible to define the associated *Gibbs measure*, given by :

$$d\mathbb{P}_{N,V,\beta,g}(x_1, \cdots, x_N) = \frac{1}{Z_{N,V,\beta,g}} \exp\left(-\beta E_N(x_1, \cdots, x_N)\right) d\pi^{\otimes N}(x_1, \cdots, x_N), \tag{8}$$

where $d\pi^{\otimes N}(x_1, \cdots, x_N) = d\pi(x_1) \cdots d\pi(x_N)$, with $\pi$ a reference measure, most of the time chosen to be the Lebesgue measure on $\mathbb{R}^d$ and $Z_{N,V,\beta,g}$ is a normalizing constant such that $\mathbb{P}_{N,V,\beta,g}$ is a probability measure[10].

Coulomb gases correspond to a particular choice of the (repulsive) interaction kernel $g$. It satisfies the so-called Poisson equation $\Delta g = -c_d \delta_0$, with $c_d$ an appropriate constant depending on the dimension $d$ so that its solution reads:

$$g(x) = \begin{cases} -|x|, & \text{for } d = 1, \\ -\log|x|, & \text{for } d = 2, \\ \frac{1}{|x|^{d-2}}, & \text{for } d \geq 3. \end{cases}$$

**Example:**  Similarly to what we saw in the first chapter of this course for the GUE, if one defines the Complex Ginibre Ensemble, as a random matrix of size $N \times N$, with independent identically distributed entries $G_{i,j}$ that are complex centered Gaussian with variance $1/N$ (without any symmetry assumption), then, one can check that the joint law of its eigenvalues is a Coulomb gas in dimension $d = 2$, with Coulomb kernel $g(x - y) = -\log|x - y|$ and quadratic external potential $V(x) = |x|^2/2$.

As mentioned earlier, the eigenvalues of the GUE do not form *stricto sensu* a Coulomb gas, but rather a so-called *log-gas* in the sense that $g(x) = -\log|x|$ although we are in dimension 1. This log-gas in one dimension is also commonly called a $\beta$-ensemble.

An important family of related particle systems are *Riesz gases*: for $d \geq 1$, $g(x) = |x|^{-s}$ with $s > 0$. We refer the reader to the survey [14] by M. Lewin.

As in the first chapter, we will study the *global regime* of these particle systems, through the first order asymptotics of the associated empirical measure

$$\hat{\mu}_N := \frac{1}{N} \sum_{i=1}^{N} \delta_{x_i}.$$

---

[10]Appropriate assumptions on $V$ and $g$ ensure in particular that $0 < Z_{N,V,\beta,g} < \infty$.

Let us now briefly mention an important topic that we will not discuss in these lectures, namely the *microscopic structure of Coulomb gases*. As we have seen in the first chapter, with the scaling that we have chosen (multiply each entry of the matrix by $1/\sqrt{N}$, or equivalently put a factor of $N$ in front of the external potential $V$ in the definition of the Hamiltonian), the weak limit of the empirical measure of the eigenvalues of the GUE is compactly supported. One can check that, under standard assumption on $V$, it would be the same for the Coulomb gas (8), associated with the Hamiltonian (7). The heuristics is that, considering a given particle $x_i$, the force $NV(x_i)$ created on it by the external potential is of the same order as the force felt from the repulsion $\sum_{j\neq i} g(x_i - x_j)$ of all the other particles, both being of order $N$: this leads to an equilibrium at a finite scale.

The limiting measure being compact, it means that on average, each particle occupies a box of volume of order $N^{-1/d}$. If one wants to study the microscopic structure of the Coulomb gas, it is therefore natural to choose a place around which there are particles, that is a point $x_0$ in the interior of the support of the limiting measure and blow up the configuration of points around $x_0$ at a scale where there would be in average one point per unit volume, that is consider the process $(N^{1/d}(x_i - x_0))_{1\leq i \leq N}$. Following the breakthrough papers by S. Serfaty and collaborators, there has been huge mathematical progresses in the study of the Coulomb gases at this new scale. One of the main features is that, similarly to what was observed for matrix models, the microscopic structure of Coulomb gases is much more universal than their global regime, in the sense that the limiting random process essentially does not depend on the external potential $V$. It does depend on $\beta$ and there is an important conjecture, that at low temperature (that is in the regime $\beta \to \infty$), there would be a *crystallization* phenomenon, the limiting process being the triangular lattice in dimension 2. We won't treat this problem in these notes but the interested reader may find a lot of resources on this topic on S. Serfaty's webpage[11]. We recommend in particular the recent survey [15].

## 2.2 General Laplace principle for particle systems driven by a $k$-body interaction

Let us now go back to our main subject and present the framework of [6], which is a very general model with a $k$-body interaction (in most physical examples, we consider pairwise interactions, that is $k = 2$). At each step, we will try to make as transparent as possible the correspondence with the GUE model studied in the first part of this course.

If $M$ is the space in which the particles live ($M$ may be $\mathbb{R}^d$, a manifold or a Polish space[12]) and $\mathcal{P}(M)$ the set of probability measures on $M$, we consider $G : M^k \to (-\infty, \infty]$, a symmetric, lower semi-continuous and bounded below function.

For $N \geq k$, we define $W_N : M^N \to (-\infty, \infty]$ by

$$W_N(x_1, \cdots, x_N) = \frac{1}{N^k} \sum_{\substack{\{i_1, \cdots, i_k\} \subseteq \{1, \cdots, N\} \\ \#\{i_1, \cdots, i_k\} = k}} G(x_{i_1}, \cdots, x_{i_k}). \tag{9}$$

For instance, for the GUE, we choose $M = \mathbb{R}$, $k = 2$ and define

$$G(x, y) := \frac{x^2}{2} + \frac{y^2}{2} - 2\log|x - y|.$$

---

[11] https://math.nyu.edu/~serfaty/

[12] A Polish space is a complete separable metric space. Working in a Polish space is a standard assumption in probability theory.

This gives

$$W_N(x_1, \cdots, x_N) = \frac{1}{N^2} \sum_{i<j} G(x_i, x_j) = \frac{1}{N^2} \left( (N-1) \sum_{i=1}^{N} \frac{x_i^2}{2} - \sum_{i \neq j} \log|x_i - x_j| \right),$$

394  which is to compare with the energy $E$ of the configuration that has been defined in (4).

395      Consider now a reference measure $\pi$ and inverse temperature $\beta_N > 0$. Similarly to what
396  we did previously, one can define an associated Gibbs measure $\gamma_N$, which has the following
397  density

$$d\gamma_N(x_1, \cdots, x_N) := \exp(-N\beta_N W_N(x_1, \cdots, x_N)) d\pi(x_1) \cdots d\pi(x_N). \qquad (10)$$

398  Note that at this stage, $\gamma_N$ is not normalized, it may not be a probability measure.
399

Again, it may be useful to compare to our example: with $G(x, y) := \frac{x^2}{2} + \frac{y^2}{2} - 2\log|x - y|$,
$\beta_N = N$ and $d\pi(x) = e^{-x^2/2} dx$, we get that

$$\mathbb{P}_{\text{GUE}_N} = C_N \gamma_N,$$

400  where $\mathbb{P}_{\text{GUE}_N}$ has been defined in (2) and $C_N$ in (3).
401      We are now ready to state the main result of [6]:

402  **Theorem 2.1** *Assume that $G : M^k \to (-\infty, \infty]$ is symmetric, lower semi-continuous and*
403  *bounded below and $W_N$ and $\gamma_N$ being defined in (9) and (10) respectively.*
404      *Assume that $\beta_N \xrightarrow[N\to\infty]{} \beta \in (0, \infty]$.*
      *Let $W : \mathcal{P}(M) \to (-\infty, \infty)$ be defined as*

$$W(\mu) := \frac{1}{k!} \int_{M^k} G(x_1, \cdots, x_k) d\mu(x_1) \cdots d\mu(x_k).$$

405      *If $\beta = \infty$, assume in addition that $G(x_1, \cdots, x_k) \xrightarrow[x_i \to \infty]{} \infty$ (i.e. we have a confining*
406  *potential) and that we have the following regularity assumption: for any $\mu \in \mathcal{P}(M)$ such that*
407  *$W(\mu) < \infty$, there exists a sequence of probability measures $(\mu_N)_{N\geq 1}$ absolutely continuous with*
408  *respect to $\pi$ such that $W(\mu_N) \xrightarrow[N\to\infty]{} W(\mu)$ as $N$ converges to $\infty$.*
409

*Then, for all $f : \mathcal{P}(M) \to \mathbb{R}$ continuous and bounded, we have*

$$\frac{1}{N\beta_N} \log \int_{M^N} \exp\left( -N\beta_N f\left( \frac{1}{N} \sum_{k=1}^{N} \delta_{x_k} \right) \right) d\gamma_N((x_1, ..., x_N) \xrightarrow[N\to\infty]{} - \inf_{\mu \in \mathcal{P}(M)} \{ f(\mu) + F(\mu) \},$$

*where $F$ is the free energy with parameter $\beta$ :*

$$F(\mu) := W(\mu) + \frac{1}{\beta} S(\mu|\pi),$$

*with $S(\cdot|\pi)$ the relative entropy (or KL divergence) :*

$$S(\mu|\pi) := \begin{cases} \int \frac{d\mu}{d\pi} \log\left( \frac{d\mu}{d\pi} \right) d\pi, & \text{if } \mu \text{ has a density with respect to } \pi, \\ \infty, & \text{otherwise.} \end{cases}$$

410      From there, one can deduce automatically an LDP for (a normalized version of) $\gamma_N$.

411  **Corollary 2.2** *Under the same assumption as in Theorem 2.1, if we define $dP_N = \frac{1}{Z_N} d\gamma_N$, where*
412  *$Z_N = \gamma_N(M^N)$, then under $P_N$, $\hat{\mu}_N = \frac{1}{N} \sum_i \delta_{x_i}$ satisfies an LDP at speed $N\beta_N$ with rate function*
413  *$J(\mu) = F(\mu) - \inf F$.*

In particular, one can recover from there the LDP in the GUE case, initially due to G. Ben Arous and A. Guionnet. In this case, as we have $k = 2$, $G(x, y) = \frac{x^2 + y^2}{2} - 2\log|x - y|$ and $\beta = \infty$, it comes that

$$W(\mu) = F(\mu) = \int \frac{x^2}{2} \mathrm{d}\mu(x) - \iint \log|x - y| \, \mathrm{d}\mu(x)\mathrm{d}\mu(y),$$

and we recover the rate function $I$ defined in (5).

Before going into more examples and then into the proof of Theorem 2.1, it is worth explaining a very general mechanism, that allows to deduce an LDP such as Corollary 2.2 from a Laplace principle as obtained in Theorem 2.1. It is an important mathematical tool in the theory of large deviations and we devote the next section to explaining this mechanism.

## 2.3   Link between Laplace principle and LDP : the Varadhan-Bryc approach

Let us first make a quick reminder on the Laplace method, which is very familiar to mathematical physicists. The *Laplace principle* states that, under suitable conditions, if we let

$$I_n := \int_{\mathbb{R}} \exp\left(n\phi(x)\right) \mathrm{d}x,$$

with $\phi$ a concave function reaching its maximum at a point $x_0$, then we should have

$$I_n \simeq \exp\left(n\phi(x_0)\right),$$

in the sense that

$$\lim_{n \to \infty} \frac{1}{n} \log I_n = \phi(x_0),$$

(one can often be more precise, depending on the regularity of the function $\phi$). In the context of large deviations, *Varadhan's lemma* can be seen as an extension of the Laplace principle: if a sequence of probability measures $\{\mu_n\}_{n \geq 1}$ defined on a space $X$, satisfies an LDP at speed $n$ with rate function $I$, and we let $J_n := \int \exp\left(n\phi(x)\right) \mathrm{d}\mu_n(x)$, then

$$\lim_{n \to \infty} \frac{1}{n} \log J_n = \sup_{x \in X}\{\phi(x) - I(x)\}.$$

One can even give a kind of reciprocal statement to Varadhan's lemma : if such a limit occurs for a rich enough family of test functions $\phi$, then an LDP holds for the sequence $\{\mu_n\}_{n \geq 1}$. This reciprocal statement is known as *Bryc's lemma*.

More precisely, we will discuss the equivalence of the two statements : for $\{\mu_n\}_{n \geq 1}$ a family of probability measures on a Polish space $X$ we consider

- (LDP) The sequence $\{\mu_n\}_{n \geq 1}$ satisfies an LDP with speed $n$, and with a good rate function $I$[13].

- (LIM) For any continuous bounded function $f$, the following limit exists

$$\Lambda_f := \lim_{n \to \infty} \frac{1}{n} \log \int \exp(nf(x))\mathrm{d}\mu_n(x).$$

The following proposition discusses the relationship between these two statements :

---

[13]We recall that by definition of semi-continuity, the level sets $\{I \leq C\}$ of rate functions are closed, when in addition these level sets are all compact then the rate function is said to be good

428 **Proposition 2.3**

429

1. *Varadhan's integral lemma: Suppose (LDP) holds then (LIM) is verified and*

$$\Lambda_f = \sup_{x \in X} \{f(x) - I(x)\}.$$

2. *Bryc's inverse integral lemma: Suppose (LIM) holds and suppose in addition that the sequence $(\mu_n)_{n \geq 1}$ is exponentially tight, then (LDP) holds with rate function $I$ defined as follows*

$$I(x) = \sup_{f \in \mathcal{C}^b} \{f(x) - \Lambda_f\},$$

430     *where $\mathcal{C}^b$ is the set of continuous bounded functions.*

431     As emphasized above, the first statement can be seen as an infinite dimensional extension
432 of Laplace method. We refer the reader to the notes of H. Touchette [1] for a more thorough
433 discussion of Varadhan's lemma in the context of statistical mechanics or to Section 4.3 of [2]
434 for a complete proof.
435     In the sequel, we will use more specifically the second statement, a.k.a. Bryc's inverse
436 integral lemma, whose proof we detail hereafter. Let us assume that (LIM) holds and that the
437 sequence $(\mu_n)_{n \geq 1}$ is exponentially tight, in the sense that there exists a sequence of compact
438 sets $(K_L)_{L \geq 0}$ such that

$$\limsup_{L \to \infty} \limsup_{n \to \infty} \frac{1}{n^2} \log \mathbb{P}(\widehat{\mu}_n \notin K_L) = -\infty.$$

We first show that the LDP lower bound holds with rate function $I$. Let $O$ be an open set
and $x \in O$, let $f$ be a bounded and continous function chosen such that, $f(x) = 1, 0 \leq f \leq 1$;
and $f = 0$ on $O^c$ (such a function can be shown to exist if $X$ is a completely regular topological
space). Then define the family of functions $(f_p)_{p \geq 1}$ by $f_p(y) = p(f(y) - 1)$, for any $y \in X$.
Thus

$$\int \exp(n f_p(y)) \mathrm{d}\mu_n(y) = \int_O \exp(n f_p) \mathrm{d}\mu_n + \int_{O^c} \exp(n f_p) \mathrm{d}\mu_n \leq \mu_n(O) + e^{-np},$$

where the inequality comes from the fact that $f_p \leq 0$ and so $\exp(n f_p) \leq 1$ and that $f_p(y) = -p$
on $O^c$. Then, taking $\liminf \frac{1}{n} \log(\cdot)$ on both sides of the previous inequality and using that

$$\liminf \frac{1}{n} \log(a_n + b_n) = \max \left\{ \liminf \frac{1}{n} \log(a_n), \liminf \frac{1}{n} \log(b_n) \right\},$$

we get :

$$\liminf_{n \to \infty} \frac{1}{n} \log \int e^{n f_p} \mathrm{d}\mu_n \leq \max \left\{ \liminf_{n \to \infty} \frac{1}{n} \log \mu_n(O), -p \right\}.$$

Given that (LIM) holds, the left hand side is $\Lambda_{f_p}$. In addition, we have $f_p(x) = 0$ so we get :

$$\Lambda_{f_p} - f_p(x) \leq \max \left\{ \liminf_{n \to \infty} \frac{1}{n} \log \mu_n(O), -p \right\}.$$

439 We therefore obtain, for any $x \in O$,

$$-I(x) := -\sup_{f \in \mathcal{C}^b} \{f(x) - \Lambda_f\} = \inf_{f \in \mathcal{C}^b} \{\Lambda_f - f(x)\} \leq \Lambda_{f_p} - f_p(x)$$

$$\leq \max \left\{ \liminf_{n \to \infty} \frac{1}{n} \log \mu_n(O), -p \right\} \xrightarrow[p \to \infty]{} \liminf_{n \to \infty} \frac{1}{n} \log \mu_n(O).$$

This inequality holds for all $x \in O$, so taking $\sup\limits_{x \in O}$ on the left hand side, we get the LDP lower bound :

$$-\inf_{x \in O} I(x) = \sup_{x \in O}\{-I(x)\} \leq \liminf_{n \to \infty} \frac{1}{n} \log \mu_n(O).$$

Let us now show the upper bound. Since we have assumed exponential tightness of $(\mu_n)_{n \geq 1}$, it is sufficient to show the upper bound for compact sets. Let $\delta > 0$, and define

$$I^\delta(x) := \min\left\{I(x) - \delta, \frac{1}{\delta}\right\}.$$

Fix a compact set $K \subset X$. By definition of $I$, for all $x \in K$, there exists $f_x \in \mathcal{C}^b(X)$ such that $f_x(x) - \Lambda_{f_x} \geq I(x) - \delta \geq I^\delta(x)$. By continuity of $f_x$, there exists an open set $A_x$ containing $x$, such that for all $y \in A_x$, $f_x(y) - f_x(x) \geq -\delta$. Now, let

$$\Lambda^{(n)}_{f_x} := \frac{1}{n} \log\left(\int \exp\left(n f_x(y)\right) d\mu_n(y)\right),$$

and define the following probability measures $\mu_{n, f_x}$ with densities :

$$d\mu_{n, f_x}(y) = \exp\left[n\left(f_x(y) - \Lambda^{(n)}_{f_x}\right)\right] d\mu_n(y).$$

Since $f_x(y) - f_x(x) \geq -\delta$ for all $y \in A_x$, we have :

$$\mu_n(A_x) = \int_{A_x} \exp\left(-n\left(f_x(y) - \Lambda^{(n)}_{f_x}\right)\right) d\mu_{n, f_x}(y) \leq \exp\left[-n\left(f_x(x) - \delta - \Lambda^{(n)}_{f_x}\right)\right].$$

Since by (LIM), $\Lambda^{(n)}_{f_x} \xrightarrow[n \to \infty]{} \Lambda_{f_x}$ and since we have chosen $f_x$ such that $f_x(x) - \Lambda_{f_x} \geq I^\delta(x)$, we get :

$$\limsup_{n \to \infty} \frac{1}{n} \log \mu_n(A_x) \leq \Lambda_{f_x} - f_x(x) + \delta \leq -I^\delta(x) + \delta.$$

By compactness of $K$, since $\bigcup_{x \in X} A_x$ covers $K$ we can extract a finite covering $K = \bigcup_{i=1}^N A_{x_i}$, for some $x_1, ..., x_N \in X$. We therefore get :

$$\limsup_{n \to \infty} \frac{1}{n} \log \mu_n(K) \leq \limsup_{n \to \infty} \frac{1}{n} \log\left(\sum_{i=1}^N \mu_n(A_{x_i})\right)$$

$$= \max_{1 \leq i \leq N}\left\{\limsup_{n \to \infty} \frac{1}{n} \log \mu_n(A_{x_i})\right\} \leq \max_{1 \leq i \leq N}\left\{-I^\delta(x_i) + \delta\right\}.$$

Taking $\lim\limits_{\delta \to 0^+}$ on the right hand side of the inequality, we obtain :

$$\limsup_{n \to \infty} \frac{1}{n} \log \mu_n(K) \leq \max_{1 \leq i \leq N}\{-I(x_i)\} \leq -\inf_{x \in X} I(x),$$

which is the upper bound for the LDP. Thus, $I$ satisfies both the upper and lower bounds, so we have an LDP with rate function $I$.

Applying Proposition 2.3, one can deduce Corollary 2.2 from Theorem 2.1.

## 2.4 Various applications of Theorem 2.1 and Corollary 2.2

The goal of the section is to exhibit several very interesting applications of the main results of [6]. We will develop some of them in details, while others will be just briefly mentioned, referring the reader to the original paper for more details.

### 2.4.1  Usual Coulomb and Riesz gases

The first result we want to establish is an LDP for the Gibbs measure $\mathbb{P}_{N,V,\beta,g}$ as defined by (8), with the kernel $g$ being a Coulomb or a Riesz kernel. On this subject, one has to mention the work of D. Chafaï, N. Gozlan and P. A. Zitt in [13] and the work of P. Dupuis, V. Laschos and K. Ramanan [16], the latter being closer in the methods of the work of D. García-Zelada. We explain hereafter how to recover those results from Theorem 2.1.

Let $\pi$ be a reference measure on $\mathbb{R}^d$. Let $V : \mathbb{R}^d \mapsto (-\infty, \infty]$ be lower semicontinuous, bounded below such that there exists $\xi > 0$ such that $\int_{\mathbb{R}^d} e^{-\xi V} d\pi < \infty$.

Let $g : \mathbb{R}^d \mapsto (-\infty, \infty]$ symmetric, lower semicontinuous such that there exists $\varepsilon > 0$ such that $(x, y) \mapsto g(x - y) + \varepsilon V(x) + \varepsilon V(y)$ is bounded below. We also assume that $(x, y) \mapsto g(x, y) + V(x) + V(y)$ goes to infinity as $x$ and $y$ both go to infinity and that the regularity assumption is satisfied.

Then, for all $f : \mathcal{P}(\mathbb{R}^d) \to \mathbb{R}$ continuous and bounded, we have

$$\frac{1}{N^2\beta} \log \int_{M^N} \exp\left(-N^2\beta f\left(\frac{1}{N}\sum_{k=1}^{N}\delta_{x_k}\right)\right) d\mathbb{P}_{N,V,\beta,g}(x_1,...,x_N) \tag{11}$$
$$\xrightarrow[N\to\infty]{} - \inf_{\mu\in\mathcal{P}(M)}\{f(\mu) + W(\mu)\},$$

and consequently, under $\mathbb{P}_{N,V,\beta,g}$, the empirical measure of the particles satisfies a large deviation principle, at speed $N^2$, with rate function $\beta J$, given by $J(\mu) = W(\mu) - \inf W$.

As already mentioned above, choosing $d = 1$, $g(x) = -2\log|x|$ and $V(x) = x^2/2$, it encompasses in particular the GUE case but also applies, with appropriate choices of the potential $V$ to any Coulomb or Riesz kernel.

Let us now quickly explain how to obtain (11) and the corresponding LDP from Theorem 2.1 and Corollary 2.2.

As we know that $\int_{\mathbb{R}^d} e^{-\xi V} d\pi < \infty$, we may assume without loss of generality that $\int_{\mathbb{R}^d} e^{-\xi V} d\pi = 1$. If we make the following choices: $\beta_N = N\beta$, and

$$G(x, y) := g(x - y) + \frac{1}{N-1}\left(N - \frac{N}{\beta_N}\xi\right)V(x) + \frac{1}{N-1}\left(N - \frac{N}{\beta_N}\xi\right)V(y),$$

with $g$ a Coulomb or Riesz kernel and $V$ an appropriate choice so that the assumptions above are satisfied[14] and define that corresponding measure $\gamma_N$ as in (10), we get

$$d\gamma_N(x_1,\ldots,x_N) = e^{-N\beta_N W_N} d\left(e^{-\xi V}\pi\right)^{\otimes N}(x_1,\ldots,x_N)$$
$$= e^{-\frac{\beta_N}{N}\left(\sum_{i<j} g(x_i - x_j) + \left(N - \frac{N}{\beta_N}\xi\right)\sum_{i=1}^{N} V(x_i)\right)} d\left(e^{-\xi V}\pi\right)^{\otimes N}(x_1,\ldots,x_N)$$
$$= Z_{N,V,\beta,g} d\mathbb{P}_{N,V,\beta,g}(x_1,\ldots,x_N),$$

which corresponds to an unnormalized version of Coulomb/Riesz gases.

We now use the following decomposition:

$$G_1(x, y) := g(x - y) + \varepsilon V(x) + \varepsilon V(y),$$

and

$$G_2(x, y) := (1 - \varepsilon)V(x) + (1 - \varepsilon)V(y),$$

---

[14]Any polynomial of even degree and positive main coefficient is suitable.

so that, with obvious notations

$$W_N = W_{N,1} + a_N W_{N,2},$$

with

$$a_N := \frac{1}{1-\varepsilon}\left(\frac{1}{N-1}\left(N - \frac{N}{\beta_N}\xi\right) - \varepsilon\right),$$

475  a sequence converging to 1 as $N$ goes to infinity. We can then check separately that $W_{N,1}$ and
476  $W_{N,2}$ satisfy the required assumptions.

477

478  As we have shown in the first chapter with Wigner's theorem, it is possible to characterize
479  the minimizer of the rate function through Euler-Lagrange equations. The minimizer is usually
480  called *equilibrium measure* and is compactly supported.

481

482  ### 2.4.2   High temperature Coulomb and Riesz gases

483  As explained in the previous subsection, the study of $\mathbb{P}_{N,V,\beta,g}$ which is related to standard
484  models in RMT corresponds to a choice of $\beta_N$ of order $N$, leading to an LDP at scale $N^2$ and
485  a limiting equilibrium measure with compact support. But the study of measures of the type
486  $\mathbb{P}_{N,V,\frac{\beta}{N},g}$ has also been considered in the literature. In this case, the corresponding particle
487  systems are for example related to the classical Toda chain [17, 18] and are often called *high*
488  *temperature $\beta$-ensembles* or *high temperature gases*. In our framework, it corresponds to a
489  choice of $\beta_N$ of order 1. This regime has been investigated by various authors, see e.g. [19, 20].
490  In this case, one can see from the definition of the function $F$ in Theorem 2.1 that the rate
491  function is a mixture of an energy term $W$ and an entropy term $S$. As far as we know, the
492  first appearance of an LDP for such particle systems goes back to the work of T. Bodineau and
493  A. Guionnet in [21] and before the work of D. García-Zelada, general results appeared in [16].
494  In the framework of [6], Laplace principle and LDP at fixed $\beta$ even require less assump-
495  tions. With the same decomposition as in Section 2.4.1, one can show the following

496  **Theorem 2.4** *Let $\pi$ be a reference measure on $\mathbb{R}^d$. Let $V : \mathbb{R}^d \mapsto (-\infty, \infty]$ lower semicontinu-*
497  *ous, bounded below such that there exists $\xi > 0$ such that $\int_{\mathbb{R}^d} e^{-\xi V} d\pi < \infty$.*
498  *    Let $g : \mathbb{R}^d \mapsto (-\infty, \infty]$ symmetric, lower semicontinuous such that there exists $\varepsilon > 0$ such*
499  *that $(x, y) \mapsto g(x - y) + \varepsilon V(x) + \varepsilon V(y)$ is bounded below.*
500  *    Then, if $\beta_N \to \beta$, for all $f : \mathcal{P}(\mathbb{R}^d) \to \mathbb{R}$ continuous and bounded, we have*

$$\frac{1}{N\beta_N}\log\int_{M^N}\exp\left(-N\beta_N f\left(\frac{1}{N}\sum_{k=1}^{N}\delta_{x_k}\right)\right)d\mathbb{P}_{N,V,\beta,g}((x_1,...,x_N)) \tag{12}$$

$$\xrightarrow[N\to\infty]{} -\inf_{\mu\in\mathcal{P}(M)}\left\{f(\mu) + W(\mu) + \frac{1}{\beta}S(\mu|\pi)\right\},$$

501  *and consequently, under $\mathbb{P}_{N,V,\beta,g}$, the empirical measure of the particles satisfies a large deviation*
502  *principle, at speed $N$, with rate function $\beta W + S(\cdot|\pi) - \inf(\beta W + S(\cdot|\pi))$.*

503  As we have shown in the first chapter with Wigner's theorem, it is possible to characterize
504  the minimizer of the rate function through Euler-Lagrange-like equations. The minimizer is
505  usually called *thermal equilibrium measure* and is not compactly supported. When this mini-
506  mizer is unique, it is again possible to deduce almost sure convergence of the empirical measure
507  to the thermal equilibrium measure.

### 2.4.3   Conditional Gibbs measures

In some cases, it may also be natural to consider a gas of $N$ particles $\{x_1, \cdots, x_N\}$ where all but the first $\ell$ points are deterministic. In [6], several regimes are considered but for the sake of simplicity, we will stick in these notes to the case when $\ell$ is of order 1.

Assume that the density of these deterministic points converges weakly to $\nu$ :

$$\nu_N := \frac{1}{N-\ell} \sum_{i=\ell}^{N} \delta_{x_i} \xrightarrow[N\to\infty]{w} \nu.$$

Define the external potential $G^E$ generated by $y$ on the random points $(x_1, \cdots, x_\ell)$ by

$$G^E((x_1, \cdots, x_\ell), y) := \sum_{i=1}^{\ell} G(x_i, y),$$

and denote the average external interaction by

$$V_N(x_1, \cdots, x_\ell) = \int G^E((x_1, \cdots, x_\ell), y) \, \mathrm{d}\nu_N(y).$$

Then, consider the internal interaction

$$G^I(x_1, \cdots, x_\ell) = \sum_{1 \le i < j \le \ell} G(x_i, x_j).$$

Finally, define $V : M^\ell \to \mathbb{R}$ by

$$V(x_1, \cdots, x_\ell) = \int_M G^E((x_1, \cdots, x_\ell), y) \, \mathrm{d}\nu(y).$$

**Theorem 2.5** *Assume that the interaction $G$ is such that the following limit holds:*

$$V(x_1, \cdots, x_\ell) = \lim_{N\to\infty} V_N(x_1, \cdots, x_\ell)$$

*and $V$ is continuous bounded on $M^\ell$. Define the conditional measure $\gamma_N^{\mathsf{c}}$ as follows:*

$$\mathrm{d}\gamma_N^{\mathsf{c}}(x_1, \cdots, x_\ell) = \exp\left\{-\beta_N \left(V_N + \frac{1}{N}G^I\right)(x_1, \cdots, x_\ell)\right\} \mathrm{d}\pi(x_1) \cdots \mathrm{d}\pi(x_\ell).$$

*Then, under some extra technical assumptions[15], for all $f \in \mathcal{C}^b(M^\ell)$, we have :*

$$\frac{1}{\beta_N} \log\left( \int_{M^\ell} \exp\{-\beta_N f(x_1, \cdots, x_\ell)\} \mathrm{d}\gamma_N^{\mathsf{c}}(x_1, \cdots, x_\ell) \right)$$

$$\xrightarrow[N\to\infty]{} -\inf\{f(x_1, \cdots, x_\ell) + V(x_1, \cdots, x_\ell)\}.$$

**Corollary 2.6** *Under the same assumptions as Theorem 2.5, the law of $(x_1, \cdots, x_\ell)$ under $\widetilde{\gamma}_N^{\mathsf{c}}$, which is the normalized version of $\gamma_N^{\mathsf{c}}$, satisfies an LDP at speed $\beta_N$ with the rate function $V - \inf V$.*

---

[15]In this paragraph, we won't be as precise as for the previous examples and refer the reader to the original paper.

We want to emphasize the change in the scaling: when the deviations of the whole empirical measure occurs at speed $N\beta_N$, the deviations of the law of this finite number of particles occur at speed $\beta_N$.

This is exactly what happens when we look at the deviations of the largest eigenvalue of the GUE (or the rightmost particle of a gas in dimension 1). When we look at the scale $e^{-N}$, all but the first particle can be considered as *frozen*, deterministic, with positions such that their limiting empirical measure is the semicircle distribution. This corresponds to taking $\ell = 1$, $G(x, y) = \frac{x^2}{2} - 2\log(|x - y|)$ and $\nu_N \xrightarrow[N\to\infty]{w} \mu_{sc}$. With this heuristics[16], we recover the result that the law of the largest eigenvalue $\lambda_1$ for the GOE model satisfies an LDP at speed $N$ with rate function $V - \inf V$, with

$$V(x) = \frac{x^2}{2} - \int \log(|x - y|)\mathrm{d}\mu_{sc}(y).$$

More details on this derivation can be found in the work [22] or in the review paper by S. Majumdar and G. Schehr [23], which presents a thorough study of the deviations of the top eigenvalue at different scales such as fluctuations, large deviations and links between the different regimes.

### 2.4.4  Further examples

We would like to finish this list of applications of the main results of [6] by mentioning two other families of particle systems that can be studied in this framework. We won't detail these examples but refer the interested reader to the original papers:

- one can recover and generalize the results of R. Berman on Coulomb gases on Riemannian manifolds, see e.g. [24],

- if we consider random polynomials on the form $P_n(z) = \sum_{k=0}^{n} a_k z^k$, where $a_k$ are i.i.d. $\mathcal{N}_{\mathbb{C}}(0, 1)$ coefficients, it is known that the zeroes form a random particle systems of Coulomb-type. The large deviations of their empirical measure have been explored, e.g. in [25] and [26]. Their results can be recovered and generalized in the framework of [6].

### 2.5  Elements of proof of the Laplace principle

We now end this chapter by giving some ideas of the proof of Theorem 2.1. We start by recalling a result on the Legendre transform of the entropy.

**Lemma 2.7** *(Legendre transform of entropy) Let $\mu$ be a probability measure on a space $E$ and $g : E \to (-\infty, +\infty]$ be a measurable and bounded below function. Then,*

$$\log\left(\int e^{-g(x)}\mathrm{d}\mu(x)\right) = -\inf_{\tau \in \mathcal{P}(E)}\left\{\int g\,\mathrm{d}\tau + S(\tau|\mu)\right\}.$$

Let give a quick proof of this lemma. If $\tau$ has a density with respect to $\mu$ and we denote by $f = \frac{d\tau}{d\mu}$ this density, recall that

$$S(\tau|\mu) = \int f\log(f)\mathrm{d}\mu = \int \log(f)\mathrm{d}\tau.$$

---

[16]The heuristics would be a rigorous application of Theorem 2.5 if all but the largest particle would be deterministic.

536  Therefore,

$$-\int g\,\mathrm{d}\tau - S(\tau|\mu) = -\int g\,\mathrm{d}\tau - \int \log(f)\mathrm{d}\tau = \int \log e^{-g}\,\mathrm{d}\tau - \int \log(f)\mathrm{d}\tau$$

$$= \int \log\big(e^{-g}/f\big)\mathrm{d}\tau \le \log\left(\int (e^{-g}/f)\mathrm{d}\tau\right) = \log\left(\int e^{-g}\,\mathrm{d}\mu\right).$$

If $\tau$ is not absolutely continuous with respect to $\mu$, then the inequality is trivially verified since in this case $S(\tau|\mu) = +\infty$. Now, taking $\sup\limits_{\tau\in\mathcal{P}(E)}$ on the left hand side we get

$$-\inf_{\tau\in\mathcal{P}(E)}\left\{\int g\,\mathrm{d}\tau + S(\tau|\mu)\right\} \le \log\left(\int e^{-g}\,\mathrm{d}\mu\right),$$

537  and this gives us one inequality.

On the other hand, if we choose the probability measure $\tau$ such that $\mathrm{d}\tau = \dfrac{e^{-g}}{\int e^{-g}\,\mathrm{d}\mu}\mathrm{d}\mu$, then, one can easily check that we have equality:

$$-\int g\,\mathrm{d}\tau - S(\tau|\mu) = \log e^{-g}\,\mathrm{d}\mu,$$

538  so that we can conclude that the inequality above is in fact an equality.

539

540  Let us now use Lemma 2.7 to show the generalized Laplace principle stated in Theorem

541  2.1.

If we apply Lemma 2.7 to $E = M^N$, $\mu = \pi^{\otimes N}$ and the test function

$$g(x_1,\cdots,x_N) := N\beta_N\left[f\left(\frac{1}{N}\sum_{i=1}^N \delta_{x_i}\right) + W_N(x_1,\cdots,x_N)\right],$$

542  we get :

$$\frac{1}{N\beta_N}\log\left(\int_{M^N}\exp\left(-N\beta_N\left[f\left(\frac{1}{N}\sum_{i=1}^N \delta_{x_i}\right) + W_N(x_1,\cdots,x_N)\right]\right)\mathrm{d}\pi(x_1)\cdots\mathrm{d}\pi(x_N)\right)$$

$$= -\inf_{\tau\in P(M^N)}\left\{\int f\left(\frac{1}{N}\sum_{i=1}^N \delta_{x_i}\right)\mathrm{d}\tau(x_1,\cdots,x_N) + \int W_N(x_1,\cdots,x_N)\mathrm{d}\tau + \frac{S(\tau|\pi^{\otimes N})}{N\beta_N}\right\}.$$

To conclude, we need to show that the right handside converges, as $N\to\infty$ towards

$$-\inf_{\mu\in\mathcal{P}(M)}\left\{f(\mu) + W(\mu) + \frac{S(\mu|\pi)}{\beta}\right\}.$$

543  We will only show the upper bound and refer the reader to the original paper, concerning the
544  lower bound, which is more technical.

545  Notice first that if we fix $\mu\in\mathcal{P}(M)$ and let $\tau_N := \mu^{\otimes N}\in\mathcal{P}(M^N)$, then

$$\limsup_{N\to\infty}\inf_{\tau\in P(M^N)}\left\{\int f\left(\frac{1}{N}\sum_{i=1}^N \delta_{x_i}\right)\mathrm{d}\tau(x_1,\cdots,x_N) + \int W_N(x_1,\cdots,x_N)\mathrm{d}\tau + \frac{S(\tau|\pi^{\otimes N})}{N\beta_N}\right\}$$

$$\le \limsup_{N\to\infty}\left(\int f\left(\frac{1}{N}\sum_{i=1}^N \delta_{x_i}\right)\mathrm{d}\tau_N(x_1,\cdots,x_N) + \int W_N(x_1,\cdots,x_N)\mathrm{d}\tau_N + \frac{S(\tau_N|\pi^{\otimes N})}{N\beta_N}\right).$$

We look at the limsup of each of the three terms. First, if $(x_1, \cdots, x_N) \sim \tau_N$, it means that the $x_i$ are i.i.d. and are distributed according to $\mu$. By the law of large numbers, we have that the law of $\frac{1}{N} \sum_{k=1}^N \delta_{x_i}$ converges weakly to $\delta_\mu$ and so that for any continuous bounded function $f$ on $\mathcal{P}(M)$ we have :

$$\int f\left(\frac{1}{N} \sum_{k=1}^N \delta_{x_i}\right) d\tau_N \xrightarrow[N\to\infty]{} \int f d\delta_\mu = f(\mu).$$

546   We now go to the second term:

$$\int W_N d\tau_N = \frac{1}{N^k} \sum_{\substack{\{i_1, \cdots, i_k\} \subseteq \{1, \cdots, N\} \\ \#\{i_1, \cdots, i_k\} = k}} \int G(x_{i_1}, \cdots, x_{i_k}) d\mu^{\otimes k}(x_{i_1}, \cdots, x_{i_k})$$

$$= \frac{1}{N^k} \binom{N}{k} \int_{M^k} G(x_1, \cdots, x_k) d\mu(x_1) \cdots d\mu(x_k) \xrightarrow[N\to\infty]{} W(\mu).$$

Finally, since

$$\frac{d\mu^{\otimes N}}{d\pi^{\otimes N}}(x_1, \cdots, x_N) = \prod_{i=1}^N \frac{d\mu}{d\pi}(x_i),$$

we have $S(\mu^{\otimes N} | \pi^{\otimes N}) = N S(\mu | \pi)$, so that

$$\frac{S(\tau_N | \pi^{\otimes N})}{N\beta_N} = \frac{S(\mu|\pi)}{\beta_N} \xrightarrow[N\to\infty]{} \frac{S(\mu|\pi)}{\beta}.$$

547   Putting these three elements together in the limit above, we get that for any $\mu \in \mathcal{P}(M)$,

$$\limsup_{N\to\infty} \inf_{\tau \in P(M^N)} \left\{ \int f\left(\frac{1}{N} \sum_{i=1}^N \delta_{x_i}\right) d\tau(x_1, \cdots, x_N) + \int W_N(x_1, \cdots, x_N) d\tau + \frac{S(\tau | \pi^{\otimes N})}{N\beta_N} \right\}$$

$$\leq f(\mu) + W(\mu) + \frac{S(\mu|\pi)}{\beta},$$

548   and we can take the infimum over the right handside.

549       We refer the reader to [6] for the proof of the reverse inequality.

550   ## 2.6   Conclusion

551   In this second chapter (corresponding to an extended version of Lectures 2 and 3), we have
552   discussed a very general result developed in [6].

553   • It allows to study the large deviations of the empirical measure of particle systems given
554       by singular Gibbs measures, encompassing a large range of applications, in particular
555       usual Coulomb gases, high temperature Coulomb gases and conditional Gibbs measures.

556   • The proof of these LDPs is based on an important mathematical tool called Bryc's inverse
557       integral lemma, that can be seen as a reciprocal to Varadhan's lemma. It allows to deduce
558       LDPs from Laplace principle.

559   • In this case, the Laplace principle is intimately linked to a dual representation of the
560       relative entropy. It leads to a rate function that is in general a mixture of an energy term
561       and a relative entropy term. In the so-called *zero temperature regime*, the entropy term
562       disappears.

- The typical behavior of the corresponding empirical measures can be described by a compactly supported equilibrium measure in the usual case (as we have seen in the first chapter with the semicircle distribution) and by a non-compactly supported (thermal) equilibrium measure in the so-called *high temperature regime*.

# 3 The use of spherical integrals to study LD of largest eigenvalues of random matrices

In the first two chapters, we have mainly dealt with the global behavior of the particle systems, encoded in their empirical measure. But, in many situations, it is also relevant to study the behavior of the extremal particles - say the rightmost particle or the largest eigenvalue. Recently, A. Guionnet and J. Husson [27] and then many co-authors [28–33] have used the so-called spherical integrals to study the large deviations of the largest eigenvalue in various models of random matrices. Although the corresponding systems of particles are no longer stricly speaking Coulomb gases, they are closely related models and we would like to present this ensemble of works in this last section. We think that the ubiquity of spherical integrals in statistical physics makes it particularly relevant for this course.

## 3.1 A general overview on the models

Let us go back to the model of the GUE, and recall that $H_N \in \mathbf{GUE}_N$ has been defined in (1.1) as follows:

$$
H_N = \begin{pmatrix} \frac{H_{1,1}}{\sqrt{N}} & & \frac{H_{i,j}}{\sqrt{N}} & \\ & \ddots & & \\ \frac{H_{i,j}^*}{\sqrt{N}} & & \ddots & \\ & & & \frac{H_{N,N}}{\sqrt{N}} \end{pmatrix},
$$

where $H_{i,i}$ are independent and identically distributed random variables with distribution $\mathcal{N}_{\mathbb{R}}(0,1)$ and for $i \leq j$, $H_{i,j}$ are independent and identically distributed random variables with distribution $\mathcal{N}_{\mathbb{C}}(0,1)$.

There are several very natural generalisations of this model.

- **$\beta$-ensembles.** We know that the joint law of the eigenvalues of the $\mathbf{GUE}_N$ is given by $\mathbb{P}_{GUE_N}$, which has been defined in (2). In the joint density, proportional to

$$
\prod_{i<j}(x_i - x_j)^2 \exp\left(-\frac{N}{2}\sum_{j=1}^{N} x_j^2\right),
$$

  if we replace the quadratic potential by a more general potential $V(x_j)$ and/or if we replace the exponent 2 in the Vandermonde term by any $\beta > 0$, the corresponding particle system is called a $\beta$-*ensemble*. Large deviations for the empirical measure and for the rightmost particle in this framework has been extensively studied and we refer to Vivo's lecture for more details. In these notes, we will focus in two other types of extension of the model.

- **Wigner matrices.** If we keep the same structure of the entries, being i.i.d., up to symmetry (the matrix has to remain Hermitian or real symmetric) but relax the Gaussianity assumption, we obtain a Wigner matrix.

  It is well known that, as soon as the entries $H_{i,j}$ are centered and $\mathbb{E}(|H_{i,j}|^2) = 1$ for $i \neq j$, Wigner's theorem holds in the sense that $\widehat{\mu}_N = \frac{1}{N}\sum_{i=1}^{N}\delta_{\lambda_i} \xrightarrow[N\to\infty]{w} \mu_{\mathrm{sc}}$, the semi-circular distribution. If moreover we have $\mathbb{E}(H_{i,j}^4) < \infty$, then $\lambda_{\max}^N \xrightarrow[N\to\infty]{} \lambda^* = 2$, which is the right edge of the support of $\mu_{\mathrm{sc}}$. The large deviations have been investigated by [27–29, 33].

601   • **Deformed models.** The $GUE_N$ distribution may also be seen as a Gaussian measure on
602      the set $\mathcal{H}_N(\mathbb{C})$ of Hermitian matrices of size $N$. A natural way to modify this measure
603      is to change its mean: choose a deterministic matrix $A_N \in \mathcal{H}_N(\mathbb{C})$ and $X_N = A_N + H_N$,
604      with $H_N \in GUE_N$. As the $GUE_N$ distribution is invariant by unitary conjugation, one can
605      consider as a further generalisation a model of the form

$$X_N = A_N + U B_N U^*,$$

606      with $U$ being distributed as the Haar measure on the orthogonal group $\mathcal{O}_N$ or the unitary
607      group $\mathcal{U}_N$. The convergence of the empirical spectral measure can be described by free
608      probability (this point will be detailed a bit further in these notes) and the behavior of
609      the largest eigenvalue has been investigated in [30–32].

610   To understand the deviations of the largest eigenvalue both for Wigner matrices and de-
611   formed models, we first need to investigate a common tool, which is interesting by itself, the
612   *spherical integrals*.

### 3.2   Spherical integrals

614   Consider $A_N, B_N$ two deterministic, real diagonal $N \times N$ matrices. Define the spherical integral
615   of $A_N$ and $B_N$ as

$$I_N(A_N, B_N) := \int e^{N \mathrm{Tr}(A_N U_N B_N U_N^*)} \mathrm{d}m_N(U_N),$$

616   with $m_N$ the Haar measure on the orthonormal group

$$\mathcal{O}_N = \{O \in \mathcal{M}_N(\mathbb{R}), O^T O = O O^T = I_N\},$$

617   or the unitary group

$$\mathcal{U}_N = \{U \in \mathcal{M}_N(\mathbb{C}), U^* U = U U^* = I_N\}.$$

618   We recall that the Haar measure is the unique probability measure which is invariant under
619   conjugation (see Appendix A for more details). According to the context, the integral $I_N$ may
620   be called *Harish Chandra integral*[17] or *Itzykson-Zuber integral* or *spherical integral*. We will use
621   this latter terminology in these notes.

622   Harish Chandra in the fifties provided explicit formulas for $I_N(A_N, B_N)$. For example, we
623   have the following formula, which holds only in the unitary case:

$$I_N(A_N, B_N) := \left( \prod_{j=1}^{N} j! \right) \frac{\det(e^{a_i b_j})_{i,j \leq N}}{\prod_{i<j}(a_i - a_j) \prod_{i<j}(b_i - b_j)},$$

624   where $(a_i)_{1 \leq i \leq N}$ and $(b_j)_{1 \leq j \leq N}$ are respectively the eigenvalues of $A_N$ and $B_N$. Unfortunately,
625   this nice closed formula is not very suitable for asymptotic analysis. Nevertheless, C. Itzykson
626   and J. B. Zuber in the physics literature [34] and then twenty years later A. Guionnet and
627   O. Zeitouni [35] on a rigorous level provided some insights on the asymptotics of $I_N$. Their
628   result takes the following form:

629   **Theorem 3.1** *If*

$$\widehat{\mu}_{A_N} = \frac{1}{N} \sum_{i=1}^{N} \delta_{\lambda_i(A_N)} \xrightarrow[N \to \infty]{w} \mu_a \ and \ \widehat{\mu}_{B_N} = \frac{1}{N} \sum_{i=1}^{N} \delta_{\lambda_i(B_N)} \xrightarrow[N \to \infty]{w} \mu_b,$$

---

[17]It is in fact a particular case of the latter.

then there exists a functional $F$ such that (under some additional technical assumptions), we have the following convergence :

$$\frac{1}{N^2} \log I_N(A_N, B_N) \xrightarrow[N \to \infty]{} F(\mu_a, \mu_b).$$

One can check that when one of the limiting measures $\mu_a$ or $\mu_b$ is trivial ($= \delta_0$), the function $F$ vanishes. This means that in this case, we are not considering the spherical integrals on the right scale. The asymptotics in the case when one of the matrices, say $A_N$ is of finite rank (fixed with $N$), has been first obtained by A. Guionnet and M. Maïda [36] in the rank one case and then by A. Guionnet and J. Husson [37] in the finite rank case (see also [38] for previous partial results). The rank one case will be particularly useful for the sequel and we present it hereafter in full details. For the sake of simplicity, we stick to the orthogonal case but the results and proofs can be easily adapted to the unitary case.

We write $A_N$ and $B_N$ under the form:

$$A_N = \begin{pmatrix} \theta & \\ & (0) \end{pmatrix}, \qquad B_N = \begin{pmatrix} b_1 & & & \\ & b_2 & & \\ & & \ddots & \\ & & & b_N \end{pmatrix}.$$

In this case, we denote by $I_N(\theta, B_N) := I_N(A_N, B_N)$. We will here restrict to the case where $\theta \geq 0$, which is useful to study the deviations of the largest eigenvalue but the very same results have been shown when $\theta \leq 0$.

Before stating the result, let us recall the following notation. Given a probability measure $\mu$ on $\mathbb{R}$ and a point $x \in R$ outside the support of $\mu$, let

$$H_\mu(x) := \int_{\mathbb{R}} \frac{1}{x - y} \mathrm{d}\mu(y).$$

Depending on the context, the functional $\mu \mapsto H_\mu$ is known as the Hilbert - or the Stieltjes - transform. We have the following:

**Theorem 3.2** *Assume that* $\widehat{\mu}_{B_N} \xrightarrow[N \to \infty]{w} \mu$, *where* $\mu$ *has a compact support. Assume also that* $\lambda_{max}(B_N) = \max_{1 \leq i \leq N} b_i \xrightarrow[N \to \infty]{} \lambda$. *Then*

$$\lim_{N \to \infty} \frac{1}{N} \log I_N(\theta, B_N) = \theta v(\theta) - \frac{1}{2} \int \log(1 + 2\theta v(\theta) - 2\theta y) \mathrm{d}\mu(y) := J(\theta, \lambda, \mu), \quad (13)$$

*where*

$$v(\theta) = \begin{cases} R_\mu(2\theta), & when \ 2\theta \leq H_{\max} \\ \lambda - \frac{1}{2\theta}, & when \ 2\theta > H_{\max} \end{cases}$$

*with*

$$H_{\max} = \lim_{x \to \lambda^+} H_\mu(x),$$

*and* $R_\mu(\eta)$ *is the unique solution of*

$$\int_{\mathbb{R}} \frac{1}{R_\mu(\eta) + \frac{1}{\eta} - y} \mathrm{d}\mu(y) = \eta$$

*such that* $R_\mu(\eta) + \frac{1}{\eta}$ *is larger or equal to* $\lambda$.

654      Note that for $\mu$ and $\lambda$ given, there is a phase transition at $2\theta = H_{\max}$. For $2\theta \leq H_{\max}$
655 (subcritical case), $v(\theta)$ and therefore $J(\theta, \lambda, \mu)$ is independent of $\lambda$ but a dependence appears
656 when $2\theta > H_{\max}$. This will play a crucial role in the tilting argument.

657      We now sketch the proof of Theorem 3.2. With $A_N$ and $B_N$ chosen as above, we have

$$I_N(\theta, B_N) = \int e^{N\theta \sum_{i=1}^N b_i O_{i1}^2} \mathrm{d}m_N(O).$$

658 In the orthogonal case we are interested in, when $(O_{i1})_{i=1}^N$ is the first column vector of a
659 matrix sampled according to the Haar measure, one can show that this vector follows the
660 uniform distribution on the sphere $\mathbb{S}^{N-1} \subset \mathbb{R}^N$. If $G$ is a standard Gaussian vector of size $N$,
661 by invariance of the standard normal distribution under orthogonal transformations, $\frac{G}{\|G\|}$ also
662 follows the uniform distribution on the sphere. Thus $(O_{i1})_{i=1}^N$ has the same distribution as $\frac{G}{\|G\|}$
663 and we can write :

$$I_N(\theta, B_N) = \mathbb{E}\left( \exp\left( N\theta \frac{\sum_{i=1}^N b_i g_i^2}{\sum_{i=1}^N g_i^2} \right) \right), \quad \text{where } G = \begin{pmatrix} g_1 \\ \vdots \\ g_N \end{pmatrix} \sim \mathcal{N}(0, \mathrm{Id}_N).$$

664 By concentration of measure phenomenon, the event

$$\mathcal{E}_N = \left\{ \left| \frac{\|G\|^2}{N} - 1 \right| \leq N^{-\kappa} \right\},$$

665 where $\kappa \in (0, 1/2)$, has very high probability for $N$ large enough. Therefore, we have the
666 following approximation:

$$I_N(\theta, B_N) = \mathbb{E}\left( e^{N\theta \frac{\sum b_i g_i^2}{\sum g_i^2}} \right) \approx \mathbb{E}\left( e^{N\theta \frac{\sum b_i g_i^2}{\sum g_i^2}} \mathbf{1}_{\mathcal{E}_N} \right).$$

667 On the event $\mathcal{E}_N$, the quantity $(\sum_{i=1}^N g_i^2 - N)$ is negligible with respect to $N$. Therefore, up to
668 a factor which is negligible with respect to $e^N$, we can write the following approximation: for
669 any $v \in \mathbb{R}$,

$$I_N(\theta, B_N) \approx \mathbb{E}\left( e^{\theta \sum b_i g_i^2 - v\theta \left( \sum g_i^2 - N \right)} \mathbf{1}_{\mathcal{E}_N} \right).$$

670 By rewritting the expectation with the density of a Gaussian vector, we get

$$I_N(\theta, B_N) \approx \frac{e^{N\theta v}}{(2\pi)^{N/2}} \int e^{\theta \sum b_i g_i^2 - v\theta \sum g_i^2 - \frac{1}{2} \sum g_i^2} \mathbf{1}_{\mathcal{E}_N} \prod_{i=1}^N \mathrm{d}g_i$$

$$= \frac{e^{N\theta v}}{(2\pi)^{N/2}} \int e^{-\frac{1}{2} \sum (1 - 2\theta b_i + 2v\theta) g_i^2} \mathbf{1}_{\mathcal{E}_N} \prod_{i=1}^N \mathrm{d}g_i.$$

671 Choosing $v$ such that $1 - 2\theta b_i + 2\theta v > 0$ for all $1 \leq i \leq N$, we identify the exponential in
672 the integral as the density of a centered normal distribution of variance $\frac{1}{1 - 2\theta b_i + 2v\theta}$. We thus
673 obtain

$$I_N(\theta, B_N) \approx \frac{e^{N\theta v}}{\prod_{i=1}^N \sqrt{1 - 2\theta b_i + 2\theta v}} \mathbb{P}_{N,v}(\mathcal{E}_N),$$

674   with $\mathbb{P}_{N,\nu}$ a Gaussian measure with covariance matrix $\Gamma = \begin{pmatrix} \frac{1}{1-2\theta b_1+2\theta\nu} & 0 & \\ 0 & \ddots & 0 \\ & 0 & \frac{1}{1-2\theta b_N+2\theta\nu} \end{pmatrix}$.

675   Therefore, bounding $\mathbb{P}_{N,\nu}(\mathcal{E}_N)$ by 1, we get :

$$I_N(\theta, B_N) \lesssim e^{N\theta\nu - \frac{1}{2}\sum\log(1-2\theta b_i+2\theta\nu)},$$

676   and thus, for any $\nu \in \mathbb{R}$ such that $1-2\theta b_i + 2\theta\nu > 0$ for all $1 \le i \le N$, we have :

$$\frac{1}{N}\log(I_N(\theta, B_N)) \lesssim \theta\nu - \frac{1}{2N}\sum_{i=1}^{N}\log(1+2\theta\nu-2\theta b_i) = \theta\nu - \frac{1}{2}\int\log(1+2\theta\nu-2\theta y)\mathrm{d}\widehat{\mu}_{B_N}(y)$$

$$\xrightarrow[N\to\infty]{} \theta\nu - \frac{1}{2}\int\log(1+2\theta\nu-2\theta y)\mathrm{d}\mu(y),$$

677   which gives us an upper bound.

678   We now compute the corresponding lower bound. We have seen that under $\mathbb{P}_{N,\nu}$, each $g_i$

679   has normal distribution with variance $\frac{1}{1-2\theta b_i+2\theta\nu}$ and so

$$\mathbb{E}_{N,\nu}\left(\frac{1}{N}\sum_{i=1}^{N}g_i^2\right) = \frac{1}{N}\sum_{i=1}^{N}\frac{1}{1-2b_i\theta+2\theta\nu}.$$

680   The equation

$$\frac{1}{N}\sum_{i=1}^{N}\frac{1}{1-2b_i\theta+2\theta\nu} = 1 \tag{14}$$

681   has a unique solution in $\nu$ such that $1-2b_i\theta+2\theta\nu > 0$ for all $1 \le i \le N$, which we denote

682   by $\nu_N(\theta)$. Thus, we get

$$\mathbb{E}_{N,\nu_N(\theta)}\left(\frac{\|G\|^2}{N}\right) = 1$$

683   and using Gaussian concentration again, we get that $\mathbb{P}_{N,\nu_N(\theta)}(\mathcal{E}_N)$ goes to 1 as $N$ grows to

684   infinity. Thus, we get that :

$$I_N(\theta, B_N) \approx e^{N\left(\theta\nu_N(\theta) - \frac{1}{2}\int\log(1-2\theta y+2\theta\nu_N(\theta))\mathrm{d}\widehat{\mu}_{B_N}(y)\right)}.$$

685   If we denote by

$$H_N(z) := H_{\widehat{\mu}_{B_N}}(z) = \frac{1}{N}\sum_{i=1}^{N}\frac{1}{z-b_i},$$

686   one can rewrite (14) as

$$H_N\left(\nu_N(\theta) + \frac{1}{2\theta}\right) = 2\theta.$$

687   One can then show that $\nu_N(\theta)$ converges to $\nu(\theta)$, where $\nu(\theta) = H_\mu^{(-1)}(2\theta) - \frac{1}{2\theta}$ if

688   $2\theta \in H_\mu([\lambda, +\infty))$ and $\nu(\theta) = \lambda - \frac{1}{2\theta}$ otherwise.

689   This concludes the proof of Theorem 3.2 which gives the full asymptotics of the spherical

690   integral in the rank one case.

691

692   The finite rank case has been treated by A. Guionnet and J. Husson [37]: if we have

693   $\lambda_1 > \dots > \lambda_k > \lambda^*$ (where we denote by $\lambda^*$ the right edge of the support of $\mu_b$) and

694  $\lambda_i(B_N) \xrightarrow[N\to\infty]{} \lambda_i, \forall i \in \{1,...,k\}$ (where $\lambda_i(B_N)$ is the $i$th largest eigenvalue of $B_N$), then
695  the logarithm of the integral is additive in the sense that:

$$\lim_{N\to\infty} \frac{1}{N}\log I_N(A_N, B_N) = \lim_{N\to\infty}\frac{1}{N}\log I_N(\theta_1, \cdots, \theta_k, B_N) = J(\theta_1, \lambda_1, \mu) + \cdots + J(\theta_k, \lambda_k, \mu),$$

696  where $J$ is the rank one limit appearing in Theorem 3.2.

697  Before going to the statements of the main results, let us make some final remarks on the
698  expression of $J$. If $H_\mu^{(-1)}$ is the inverse of the function $H_\mu$ on $[\lambda, \infty)$ and we denote by

$$R_\mu(z) = H_\mu^{(-1)}(z) - \frac{1}{z}$$

699  on this interval, $R_\mu$ is known by mathematicians as the $R$-transform of the measure $\mu$ and by
700  physicists as the *blue function*[18] (see e.g. J.P. Bouchaud's lecture notes [39] from les Houches
701  2015). This functional is very useful to describe the limiting spectrum of $A_N + UB_NU^*$ in our
702  model. It is a central tool in free probability theory (see for example the book of J. Mingo
703  and R. Speicher [40] for a thorough but gentle introduction to the theory). If we choose the
704  sequences $(A_N)_{N\geq 1}$ and $(B_N)_{N\geq 1}$ such that $\widehat{\mu}_{A_N} \xrightarrow[N\to\infty]{w} \mu_a$ and $\widehat{\mu}_{B_N} \xrightarrow[N\to\infty]{w} \mu_b$, one can show
705  that

$$\widehat{\mu}_{A_N + UB_NU^*} \xrightarrow[N\to\infty]{w} \mu_S,$$

706  which is characterized by the functional equation

$$R_{\mu_S}(z) = R_{\mu_a}(z) + R_{\mu_b}(z).$$

707  This relation plays the role of the additivity of the logarithm of the Fourier transform for
708  the usual convolution: if $X$ and $Y$ are independent real random variables, with respective
709  distributions $\mu_X$ and $\mu_Y$ and if $\phi_\mu$ is the characteristic function of a probability meausre $\mu$, we
710  have

$$\log F_{\mu_{X+Y}} = \log F_{\mu_X} + \log F_{\mu_Y}.$$

711  By analogy, $\mu_S$ is called the *free convolution* of $\mu_a$ and $\mu_b$ and is denoted by $\mu_S = \mu_a \boxplus \mu_b$.

## 3.3   Statement of the results

713  We will now provide a statement of the LDP for the largest eigenvalue in two different models
714  that are both a generalisation of the GUE.

**Sub-Gaussian Wigner matrices**   Let us present hereafter a result due to N. Cook, R. Ducatez
and A. Guionnet [33]; it is the outcome of a series of works, starting from [27]. For a probability measure $\mu \in \mathcal{P}(\mathbb{R})$, its log-Laplace transform is given by $\Lambda_\mu(t) = \log \int e^{tx}\,\mathrm{d}\mu(x)$. For
the standard Gaussian measure, with density :

$$\mathrm{d}\gamma(x) = \frac{1}{\sqrt{2\pi}}e^{-x^2/2}\mathrm{d}x,$$

715  one can check that its log-Laplace transform is given by $\Lambda_\gamma(t) = t^2/2$, for any $t \in \mathbb{R}$. Accord-
716  ingly, a measure $\mu$ is said to be *sub-Gaussian* if there exists $K > 0$ such that $\Lambda_\mu(t) \leq Kt^2, \forall t \in \mathbb{R}$.
717  It is said to be *sharp sub-Gaussian* if in addition $K = 1/2$.

We also recall that the rate function for the largest eigenvalue $\lambda_1$ of the GOE is given by :

$$I^\gamma(x) = \begin{cases} \frac{1}{2}\int_2^x \sqrt{y^2-4}\,\mathrm{d}y, & \text{for } x \geq 2, \\ \infty, & \text{for } x < 2. \end{cases}$$

---

[18]as it is the inverse of the Green function (sic!)

718     Let $(X_{i,j})_{1 \le i \le j \le N}$ be i.i.d. real centered random variables, with unit variance. A *Wigner*
719     *matrix* $W_N$ is defined as follows:

$$
W_N = \frac{1}{\sqrt{N}} \begin{pmatrix} \sqrt{2}X_{1,1} & & X_{i,j} & \\ & \ddots & & \\ X_{j,i} & & \ddots & \\ & & & \sqrt{2}X_{N,N} \end{pmatrix}.
$$

720     We denote by $\mu$ the common distribution of the entries. The matrix $W_N$ is said to be a *sub-*
721     *Gaussian Wigner matrix* if the distribution $\mu$ is sub-Gaussian. The large deviations of its largest
722     eigenvalue are described by the following result:

**Theorem 3.3** *For $W_N$ a sub-Gaussian Wigner matrix, the law of $\lambda_1(W_N)$ satisfies an LDP at*
724     *speed $N$, with good rate function $I^\mu$ such that*

725     • $I^\mu \le I^\gamma$;

726     • $\exists x_\mu > 2$ *such that* $I^\mu = I^\gamma$ *on* $[2, x_\mu]$ *and* $I^\mu < I^\gamma$ *if* $x > x_\mu$;

727     • $x_\mu < \infty$ *if and only if* $K > 1/2$.

728     One can also mention a few previous results, dealing with Wigner matrices with non Gaus-
729     sian tails [41], or sparse Wigner matrices [42–44].

**Orthogonally invariant deformed random matrices**    We state hereafter a similar result that
731     was obtained in [32]. The tilting argument is a bit easier to present in this case and this is why
732     we choose to emphasize this model.
733     If the distribution of $O$ is the Haar measure on the orthogonal group[19] $\mathcal{O}_N$ and $A_N$ and $B_N$
734     are deterministic diagonal[20] matrices, we define

$$
H_N := A_N + O B_N O^*.
$$

Assume that $\hat{\mu}_{A_N} \xrightarrow[N \to \infty]{w} \mu_a$, $\hat{\mu}_{B_N} \xrightarrow[N \to \infty]{w} \mu_b$, which are compactly supported, and assume that
$\lambda_1(A_N) \xrightarrow[N \to \infty]{} \rho_a$, $\lambda_1(B_N) \xrightarrow[N \to \infty]{} \rho_b$, which are the right edges[21] of $\mu_a$ and $\mu_b$ respectively.
As we have previously mentioned, we know that

$$
\hat{\mu}_{H_N} \xrightarrow[N \to \infty]{w} \mu_a \boxplus \mu_b
$$

735     and denote by $\rho(\mu_a \boxplus \mu_b)$ the right edge of the support of $\mu_a \boxplus \mu_b$. We then have the following
736     LDP:

**Theorem 3.4** *With $H_N$ defined as above, the law of its largest eigenvalue $\lambda_1(H_N)$ satisfies an LDP*
738     *at speed $N$ with good rate function $L_{a,b}$:*

$$
L_{a,b}(x) = \begin{cases} \sup_\theta L_{a,b}(\theta, x), & \text{if } x \ge \rho(\mu_A \boxplus \mu_B), \\ +\infty, & \text{if } x < \rho(\mu_A \boxplus \mu_B), \end{cases}
$$

739     *with*

$$
L_{a,b}(\theta, x) := J(\theta, x, \mu_a \boxplus \mu_b) - J(\theta, x, \mu_a) - J(\theta, x, \mu_b). \tag{15}
$$

740     *and $J$ defined in equation (13).*

---

[19]The unitarily invariant case can be treated similarly by replacing the orthogonal group by the unitary group in the sequel.

[20]$A_N$ and $B_N$ may be considered real symmetric. By invariance of the Haar measure under unitary conjugation, one can assume without loss of generality that they are diagonal.

[21]More general results are given in [32].

### 3.4 Main ideas of the proofs

In this section, we provide the main ideas of the proofs of Theorems 3.3 and 3.4. As announced, it is based on tilting the measure thanks to spherical integrals. We start by recalling how such a tilting argument has been used in the much simpler context of real i.i.d. real random variables to prove Cramér's theorem. We then show how it can be applied in our case for studying the deformed model. The sub-Gaussian Wigner case is much more involved and will only be sketched in the last paragraph.

#### 3.4.1 Tilting for Cramér

Consider $(X_N)_{N \geq 1}$ a sequence of i.i.d. real random variables that are centered, with law $\mu$ and such that the log-Laplace transform satisfies $\Lambda_\mu(t) < \infty, \forall t \in \mathbb{R}$.
Cramér's theorem states that the law of $\bar{X}_N = (X_1 + \ldots + X_N)/N$ satisfies an LDP with rate function $\Lambda_\mu^*$ defined as $\Lambda_\mu^*(x) = \sup_{\theta \in \mathbb{R}} (\theta x - \Lambda_\mu(\theta))$, for any $x \in \mathbb{R}$. This is a classical result and we refer the reader for example to [2].

The idea of the proof goes as follows : for any $x \in \mathbb{R}$ and $\delta \geq 0$,

$$\mathbb{P}\left(\left|\bar{X}_N - x\right| \leq \delta\right) = \mathbb{E}\left(\frac{e^{N\theta\bar{X}_N}}{e^{N\theta\bar{X}_N}}1_{|\bar{X}_N - x| \leq \delta}\right) \simeq e^{-N\theta x} \underbrace{\frac{\mathbb{E}\left(e^{N\theta\bar{X}_N}1_{|\bar{X}_N - x| \leq \delta}\right)}{\mathbb{E}\left(e^{N\theta\bar{X}_N}\right)}}_{=:\mathbb{P}_N^\theta(|\bar{X}_N - x| \leq \delta)} \times \underbrace{\mathbb{E}\left(e^{N\theta\bar{X}_N}\right)}_{=e^{N\Lambda_\mu(\theta)}},$$

where $\mathbb{P}_N^\theta$ is the tilted measure defined by:

$$\mathbb{P}_N^\theta(A) = \frac{\mathbb{E}\left(e^{N\theta\bar{X}_N}1_A\right)}{\mathbb{E}\left(e^{N\theta\bar{X}_N}\right)}.$$

Thus, we have:

$$\mathbb{P}\left(\left|\bar{X}_N - x\right| \leq \delta\right) \simeq e^{-N(\theta x - \Lambda_\mu(\theta))}\mathbb{P}_N^\theta\left(\left|\bar{X}_N - x\right| \leq \delta\right) \leq e^{-N(\theta x - \Lambda_\mu(\theta))},$$

and by optimizing over $\theta$, we obtain $\mathbb{P}\left(\left|\bar{X}_N - x\right| \leq \delta\right) \leq e^{-N\Lambda_\mu^*(x)}$, which is the upper bound we expect for Cramér's theorem.

On the other hand, to get a lower bound, we need to find $\theta_x$ such that $\mathbb{P}_N^{\theta_x}\left(\left|\bar{X}_N - x\right| \leq \delta\right) \geq \frac{1}{2}$. Otherwise stated, under $\mathbb{P}_N^{\theta_x}$, $x$ should be the typical behavior of $\bar{X}_N$. Now, as $\mathbb{P}_N^{\theta_x}$ preserves the independence of $X_1, \ldots, X_N$, by the law of large number, the typical value of $\bar{X}_N$ under $\mathbb{P}_N^{\theta_x}$ should be $\mathbb{E}_N^{\theta_x}(\bar{X}_N)$. By differentiating $\Lambda_\mu$, we get $\Lambda_\mu'(\theta) = \mathbb{E}_N^\theta(\bar{X}_N)$. This leads us to choose $\theta_x$ such that $\Lambda_\mu'(\theta_x) = x$. By the law of large numbers, for large enough $N$, one has $\mathbb{P}_N^{\theta_x}\left(\left|\bar{X}_N - x\right| \leq \delta\right) \geq \frac{1}{2}$. In addition, since $\Lambda_\mu'(\theta_x) = x$, we get by optimizing $\theta x - \Lambda_\mu(\theta)$ over $\theta$ that $\theta_x x - \Lambda_\mu(\theta_x) = \sup_\theta\{\theta x - \Lambda_\mu(\theta)\} = \Lambda_\mu^*(x)$ so we get the lower bound and conclude the proof.

#### 3.4.2 Tilting for $\lambda_1(H_N)$

We now go to the proof of Theorem 3.4, studying the deviations of $\lambda_1(H_N)$, with

$$H_N = A_N + OB_NO^*.$$

Mimicking the previous situation, one could try to tilt the measure directly by $e^{N\theta\lambda_1(H_N)}$. This is not a reasonable strategy as we do not know how to evaluate $\mathbb{E}(e^{N\theta\lambda_1(H_N)})$ to start with. A better strategy, relying on spherical integrals, has emerged from discussions between A. Guionnet and M. Potters. On our model, this goes as follows.

772    If we denote by $\mu = \mu_A \boxplus \mu_B$ and $\lambda_1 = \lambda_1(H_N)$, we have :

$$\mathbb{P}\left(|\lambda_1 - x| \leq \delta\right) = \mathbb{E}\left(\frac{I_N(\theta, H_N)}{I_N(\theta, H_N)} \times 1_{|\lambda_1 - x| \leq \delta}\right) \simeq \mathbb{E}\left(\frac{I_N(\theta, H_N)}{I_N(\theta, H_N)} \times 1_{|\lambda_1 - x| \leq \delta} \times 1_{\hat{\mu}_{H_N} \in B(\mu, N^{-1/4})}\right)$$

$$\simeq \exp\left(-NJ(\theta, x, \mu)\right) \mathbb{E}\left[I_N(\theta, H_N) \times \mathbb{I}_{|\lambda_1 - x| \leq \delta} \times 1_{\hat{\mu}_{H_N} \in B(\mu, N^{-1/4})}\right].$$

773    The idea behind the first approximation is that the concentration of $\hat{\mu}_{H_N}$ around $\mu$ is much
774  more robust and fast than the convergence of $\lambda_1$. This is essentially because the scaling in the
775  LDP for $\hat{\mu}_{H_N}$ is of order $N^2$, whereas that of $\lambda_1$ is of order $N$. The second approximation is
776  obtained by using that $\frac{1}{N} \log I_N(\theta, H_N)$ converges to $J(\theta, x, \mu)$ whenever $\hat{\mu}_N \simeq \mu$ and $\lambda_1 \simeq x$.
777    Now, if we define our tilting measure as:

$$\mathbb{P}_N^\theta(A) = \frac{\mathbb{E}\left(I_N(\theta, H_N) \times 1_A\right)}{\mathbb{E}\left(I_N(\theta, H_N)\right)}, \tag{16}$$

778  we get :

$$\mathbb{P}(|\lambda_1 - x| \leq \delta) \tag{17}$$
$$\simeq \exp\left(-NJ(\theta, x, \mu)\right) \times \mathbb{E}\left(I_N(\theta, H_N)\right) \times \mathbb{P}_N^\theta\left(|\lambda_1 - x| \leq \delta, \hat{\mu}_{H_N} \in B(\mu, N^{-1/4})\right).$$

779    To proceed with the tilting argument we used in the case of i.i.d. variables (as shown
780  above), we are faced with two challenges :

781    1. to get an upper bound for the LDP, we want to compute the annealed spherical integral
782       $\mathbb{E}\left(I_N(\theta, H_N)\right)$,

    2. to get a lower bound, we want to find a parameter $\theta_x$ such that

$$\mathbb{P}_N^{\theta_x}\left(|\lambda_1 - x| \leq \delta, \hat{\mu}_{H_N} \in B(\mu, N^{-1/4})\right) \geq \frac{1}{2}.$$

783    Let us start by computing the annealed spherical integral. We recall that $H_N = A_N + OB_N O^*$;
784  if we denote by $C_N = \begin{pmatrix} \theta & \\ & (0) \end{pmatrix}$ and consider $O$ and $V$ that are independent and both Haar
785  distributed on $\mathcal{O}_N$, then

$$\mathbb{E}\left(I_N(\theta, H_N)\right) = \mathbb{E}_O\left[\mathbb{E}_V\left(e^{N \operatorname{Tr}(C_N V H_N V^*)}\right)\right] = \mathbb{E}_O \mathbb{E}_V\left(e^{N \operatorname{Tr}(C_N V(A_N + OB_N O^*)V^*)}\right)$$
$$= \mathbb{E}_O \mathbb{E}_V\left(e^{N \operatorname{Tr}(C_N V A_N V^*)} e^{N \operatorname{Tr}(C_N (VO)B_N(VO)^*)}\right).$$

786  Now, as $V$ and $VO$ are also independent and Haar distributed, we end up with

$$\mathbb{E}\left(I_N(\theta, H_N)\right) = I_N(\theta, A_N) I_N(\theta, B_N). \tag{18}$$

787    This immediately gives the following upper bound:

$$\mathbb{P}(|\lambda_1 - x| \leq \delta) \leq \exp(-NJ(\theta, x, \mu)) \times \mathbb{E}(I_N(\theta, H_N))$$
$$= \exp(-NJ(\theta, x, \mu)) I_N(\theta, A_N) I_N(\theta, B_N)$$
$$\leq \exp\left\{-N\left[J(\theta, x, \mu_a \boxplus \mu_b) - J(\theta, x, \mu_a) - J(\theta, x, \mu_b)\right]\right\}$$

788  and we conclude by optimizing on $\theta$.
789

790    To get a lower bound, we want to find a parameter $\theta_x$ such that $\mathbb{P}_N^{\theta_x}(|\lambda_1 - x| \leq \delta) \geq \frac{1}{2}$.
791  As previously, we first have to understand what is the typical value of $\lambda_1(H_N)$ under $\mathbb{P}_N^\theta$. The

trick is to establish a large deviation upper bound for $\lambda_1$ under $\mathbb{P}_N^\theta$. And to use the fact that the typical value under the tilted measure will be the minimizer of the large deviation upper bound under $\mathbb{P}_N^\theta$. Using the definition of the tilted measure given in (16) and the relation obtained in (18), we have :

$$
\mathbb{P}_N^\theta(|\lambda_1 - x| \le \delta) \approx \frac{1}{I_N(\theta, A_N) I_N(\theta, B_N)} \mathbb{E}\left[ I_N(\theta, H) \mathbb{1}_{\{|\lambda_1 - x| \le \delta, \hat{\mu}_N \simeq \mu\}} \frac{I_N(\theta', H)}{I_N(\theta', H)} \right]
$$

$$
\le \frac{1}{I_N(\theta, A_N) I_N(\theta, B_N)} \sup_{H \in \mathcal{E}_N(x)} \left\{ \frac{I(\theta, H)}{I(\theta', H)} \right\} \times P_N^{\theta'}(\mathcal{E}_N(x)) \times \mathbb{E}\left( I_N(\theta', H) \right),
$$

where $\mathcal{E}_N(x) = \{|\lambda_1 - x| \le \delta\} \cap \{\hat{\mu}_N \simeq \mu\}$. We can always bound $\mathbb{P}_N^{\theta'}(\mathcal{E}_N(x))$ by 1 and by definition of $J(\theta, x, \mu)$ and the fact that on $\mathcal{E}_N(x)$ we have $\lambda_1 \simeq x$ and $\hat{\mu}_N \simeq \mu$, we also get the approximation :

$$
\sup_{H \in \mathcal{E}_N(x)} \left\{ \frac{I(\theta, H)}{I(\theta', H)} \right\} \simeq \exp\left\{ -N\left[ J(\theta, x, \mu_a \boxplus \mu_a) - J(\theta', x, \mu_a \boxplus \mu_b) \right] \right\} .
$$

Otherwise stated, with $L_{a,b}(\theta, x)$ as defined in (15), we get the following upper bound:

$$
\limsup_{N \to} \frac{1}{N} \log \mathbb{P}_N^\theta(|\lambda_1 - x| \le \delta) \le -(L_{a,b}(\theta, x) - \inf_{\theta' \ge 0} L_{a,b}(\theta', x)).
$$

A thorough study of the function $L_{a,b}$ shows that, under our assumptions on the model, there exists a unique $\theta_x$ such that $L_{a,b}(\theta_x, x) = \inf_{\theta' \ge 0} L_{a,b}(\theta', x)$ and for any $y \ne x$, we have $\inf_{\theta \ge 0} L_{a,b}(\theta, x) < L_{a,b}(\theta_x, y)$. This implies that, with this choice for $\theta_x$, we have $\mathbb{P}_N^{\theta_x}(|\lambda_1 - x| \le \delta) \ge \frac{1}{2}$ and concludes the proof of the lower bound.

### 3.4.3 Tilting for $\lambda_1(W_N)$

In the case of sub-Gaussian Wigner matrices, the very same strategy is applied but the two main technical steps, that is the computation of the annealed spherical integral and the understanding of the typical behavior of $\lambda_1$ under the tilted measures are both much more involved than in the previous case. We present here the arguments of [27] under the stronger assumption of sharp sub-Gaussianity of the entries (that is $K = 1/2$). As mentioned in the introduction of this chapter, this assumption has been progressively relaxed along a series of papers outcoming to [33], at the price of highly technical arguments that are out of the scope of these notes.

In the case of sub-Gaussian Wigner matrices, the empirical spectral measure of $W_N$ concentrates very quickly around the semi-circular distribution, that we denote again by $\mu_{\text{sc}}$. Therefore,

$$
\mathbb{P}(|\lambda_1 - x| \le \delta) = \mathbb{E}\left( \frac{I_N(\theta, W_N)}{I_N(\theta, W_N)} \times 1_{|\lambda_1 - x| \le \delta} \right) \simeq \mathbb{E}\left( \frac{I_N(\theta, W_N)}{I_N(\theta, W_N)} \times 1_{|\lambda_1 - x| \le \delta} \times 1_{\hat{\mu}_{W_N} \in B(\mu_{\text{sc}}, N^{-1/4})} \right)
$$

$$
\simeq \exp\left( -NJ(\theta, x, \mu_{\text{sc}}) \right) \mathbb{E}\left[ I_N(\theta, W_N) \times 1_{|\lambda_1 - x| \le \delta} \right].
$$

In this case, we have to consider not only one tilted measure for each $\theta \ge 0$ but a whole family of tilted measure. More precisely, if we denote by $d\nu$ the uniform measure on the unit sphere $\mathbb{S}^{N-1} \subset \mathbb{R}^N$, we write

$$
\mathbb{P}(|\lambda_1 - x| \le \delta) \simeq \exp\left( -NJ(\theta, x, \mu_{\text{sc}}) \right) \int_{\mathbb{S}^{N-1}} \mathbb{E}\left( e^{N\theta \langle \nu, W_N \nu \rangle} 1_{|\lambda_1 - x| \le \delta} \right) d\nu
$$

$$
\simeq \exp\left( -NJ(\theta, x, \mu_{\text{sc}}) \right) \int_{\mathbb{S}^{N-1}} \mathbb{E}\left( e^{N\theta \langle \nu, W_N \nu \rangle} \right) \mathbb{P}_N^{(\theta, \nu)}(|\lambda_1 - x| \le \delta) d\nu, \quad (19)
$$

815  with

$$\mathbb{P}_N^{(\theta,v)}(A) := \frac{\mathbb{E}\left(e^{N\theta\langle v, W_N v\rangle} 1_A\right)}{\mathbb{E}\left(e^{N\theta\langle v, W_N v\rangle}\right)}.$$

816  To get an upper bound, for each $\theta \geq 0$ and $v \in \mathbb{S}^{N-1}$, we need an upper bound on the
817  annealed spherical integral $\mathbb{E}\left(e^{N\theta\langle v, W_N v\rangle}\right)$, where the expectation is over the distribution of
818  $W_N$. This is provided by the following computation:

$$
\begin{aligned}
\mathbb{E}\{\exp(N\theta\langle v, W_N v\rangle)\} &= \mathbb{E}\left\{\exp\left(\theta\sqrt{N}\left[2\sum_{i<j}X_{ij}v_i v_j + \sum_i X_{ii}v_i^2\right]\right)\right\} \\
&= \exp\left\{\sum_{i<j}\Lambda_\mu(2\theta\sqrt{N}v_i v_j) + \sum_i \Lambda_\mu(\theta v_i^2\sqrt{N})\right\} \\
&\leq \exp\left\{\sum_{i<j}2\theta^2.Nv_i^2 v_j^2 + \sum_i \theta^2 N v_i^4\right\} \\
&= \exp\left\{N\theta^2\left(\sum_i v_i^2\right)^2\right\} = \exp\left(N\theta^2\right),
\end{aligned}
\tag{20}
$$

819  where we have used sharp sub-Gaussianity for the first inequality and the fact that $v \in \mathbb{S}^{N-1}$
820  for the last equality.
821  By using that $P_N^{(\theta,v)}(|\lambda_1 - x| \leq \delta) \leq 1$ in (19), and using the bound on $\mathbb{E}\{\exp(N\theta\langle v, W_N v\rangle)\}$
822  found in (20) and optimizing over $\theta \geq 0$, one gets that

$$\limsup_{N\to\infty}\frac{1}{N}\log\left(\mathbb{P}(|\lambda_1 - x| \leq \delta)\right) \leq -\inf_{\theta\geq 0}\{\theta^2 - J(x, \theta, \mu_{\mathrm{sc}})\}.$$

823  One can check that, for $x \geq 2$, the infimum is reached at $\theta_x := \frac{1}{4}(x - \sqrt{x^2 - 4})$ and equals
824  $-I^\gamma(x)$.
825  Towards the lower bound, let us now try to understand the behavior of $\lambda_1$ under $\mathbb{P}_N^{(\theta,v)}$.
826  One can check that

$$\mathbb{E}_N^{(\theta,v)}((W_N)_{i,j}) = \sqrt{\frac{1}{N}}\Lambda_\mu'\left(2\theta\sqrt{N}v_i v_j\right).$$

827  For the lower bound, the idea is that it is possible to restrict ourselves to delocalized eigen-
828  vectors $v$. Indeed, if the vector $v$ is delocalized, then the product $v_i v_j$ is much smaller than
829  $N^{-1/2}$, so that $2\theta\sqrt{N}v_i v_j = o(1)$. Now, in the vicinity of 0, we have that $\Lambda_\mu'(t) \simeq t$ so that

$$\mathbb{E}_N^{(\theta,v)}((W_N)_{i,j}) \simeq 2\theta v_i v_j.$$

830  More precisely, one can show that, if $v$ is delocalized, then under $\mathbb{P}_N^{(\theta,v)}$, we have

$$W_N \simeq \widetilde{W}_N + 2\theta v v^{\mathrm{T}},$$

831  where $\widetilde{W}_N$ is a Wigner matrix under $\mathbb{P}_N^{(\theta,v)}$. It means that $W_N$ is a rank one deformation of a
832  Wigner matrix. Such deformed models have been extensively studied (see for example [45])
833  and we know that, for $\theta \geq 2$, the typical value of $\lambda_1$ is $2\theta + \frac{1}{2\theta}$. Therefore, to get the lower
834  bound, we are lead to choose $\theta_x$ such that $2\theta_x + \frac{1}{2\theta_x} = x$. Note that this coincides with the
835  value of $\theta_x$ optimizing the upper bound. This concludes our sketch of proof of Theorem 3.3
836  in the sharp sub-Gaussian case.

## 3.5   Conclusion

In this third chapter, corresponding to an extended version of Lectures 4 and 5, we have pre-sented a general method, introduced in [27] and developed in a long series of papers to study large deviations at the edge of some random matrix models.

- We get a large deviation principle for the largest eigenvalue for sub-Gaussian Wigner matrices and for a deformation of a unitarily invariant model.

- The proof of these results uses spherical integrals, that are well known in physics and interesting mathematical objects by themselves. We have stated and proved in details their asymptotics in the case when one of the matrices is of rank one.

- The proofs also rely on a clever use of a tilting argument, which is classical in the frame-work of large deviation theory and that we have also presented in the easy case of Cramér's theorem.

## A  On Haar measures and the distribution of eigenvectors of a GUE matrix

Let $H_N$ be a random matrix in $\mathcal{H}_N(\mathbb{C})$ with distribution $\mathbb{P}_{GUE_N}$, as defined in Proposition 1.2. Any realisation $H_N(\omega)$ is Hermitian, so the matrix $U_N(\omega)$ of its eigenvectors can be chosen unitary: it belongs to

$$\mathcal{U}_N := \{ U \in \mathcal{M}_N(\mathbb{C}), U U^* = U^* U = I_N \}.$$

From the definition of $\mathbb{P}_{GUE_N}$, it is easy to check that if $H_N$ has distribution $\mathbb{P}_{GUE_N}$, then for any fixed matrix $V \in \mathcal{U}_N$, $V H_N V^*$ has the same distribution $\mathbb{P}_{GUE_N}$. Therefore, $V U_N$ has the same distribution as $U_N$. This is enough to characterize the distribution of $U_N$.

Indeed, we have the following:

**Proposition A.1** *Let $G$ be a compact topological group. There exists a unique probability measure $\mu_{Haar,G}$ that is left translation invariant i.e. $\mu_{Haar,G}(g \cdot A) = \mu_{Haar,G}(A)$, for any $g \in G$ and any Borelian subset $A \subseteq G$. This measure is called the Haar measure of the group $G$. Note that this measure is also right invariant i.e. $\mu_{Haar,G}(A \cdot g) = \mu_{Haar,G}(A)$. It is therefore also conjugation invariant.*

Heuristically, one can view the sampling according to the Haar measure of $G$ as picking a point at random and uniformly on $G$.

The group of unitary matrices $\mathcal{U}_N$ is a compact topological group and we can thus deduce from the above discussion that the distribution of the matrix $U_N$ of the eigenvectors of $H_N$ is the Haar measure on $\mathcal{U}_N$.

As a by product of the proof of the Weyl formula (2), one can also check that $U_N$ can be chosen independent of the eigenvalues $(\lambda_1^N, ..., \lambda_N^N)$. This leads to a third possible description of the GUE. To construct $H_N$, pick $U$ according to the Haar measure on the group of unitary matrices $\mathcal{U}_N$. Then, sample independently $(\lambda_1^N, ..., \lambda_N^N)$ from $\mathbb{P}_{GUE_N}$ and define $H_N := U_N \Lambda_N U_N^*$, with $\Lambda_N$ the diagonal matrix with diagonal entries $(\lambda_1^N, ..., \lambda_N^N)$.

## B  On Euler-Lagrange equations for the quadratic potential

The object of this appendix is to give a proof of Lemma 1.8.

For any $x \in \mathbb{R}$, we denote by

$$F(x) := \int \log|x - y| d\mu_{\mathrm{sc}}(y),$$

the logarithmic potential of the semicircular distribution $\mu_{\mathrm{sc}}$. Our task is to compute this quantity in two different regimes : when $x \in [-2, 2]$, that is when $x$ belongs to the support of $\mu_{\mathrm{sc}}$, which corresponds to the first equality in Lemma 1.8 and when $x \notin [-2, 2]$, that is when $x$ is outside the support, which corresponds to the second inequality.

Let us start with the first case. As an intermediate step, we compute the Stietljes transform

$$s(z) := \int \frac{1}{y - z} d\mu_{\mathrm{sc}}(y),$$

for any $z \notin \mathbb{R}$. By a simple change of variables $y = 2\cos\theta$, we can rewrite

$$s(z) = \frac{1}{\pi} \int_0^{2\pi} \frac{(\sin\theta)^2}{2\cos\theta - z} d\theta.$$

If we denote by $\xi = e^{i\theta}$, we can write it as a contour integral

$$s(z) = -\frac{1}{4i\pi} \oint_{|\xi|=1} \frac{(\xi^2-1)^2}{\xi^2(\xi^2+1-z\xi)} d\xi.$$

The poles are $\xi_0 = 0$, $\xi_1 = \frac{z+\sqrt{z^2-4}}{2}$ and $\xi_2 = \frac{z-\sqrt{z^2-4}}{2}$, where we choose the branch of the square root with positive imaginary part. One can check that $\xi_1$ is outside the unit circle and $\xi_2$ inside. Computing the residues, we have

$$\text{Res}(\xi_0) = z, \quad \text{Res}(\xi_2) = -\sqrt{z^2-4},$$

from which we get that

$$s(z) = \frac{-z+\sqrt{z^2-4}}{2}.$$

Then, $\forall x \in [-2, 2]$,

$$F'(x) = -\text{PV} \int \frac{1}{x-y} d\mu_{\text{sc}}(y) = -\lim_{\varepsilon \to 0} \int_{|y-x| \geq \varepsilon} \frac{1}{x-y} d\mu_{\text{sc}}(y) = -\frac{1}{2}(s(x+i0)+s(x-i0)) = \frac{x}{2}.$$

From there, one can deduce that

$$F(x) = \frac{x^2}{4} + C.$$

The constant $C$ will be determined by the next computation.

We now go to the case when $x \notin [-2, 2]$. By symmetry, one can assume that $x \geq 2$. From Vivo's lecture notes, Section IV.A.1., we get that

$$L(x) := \frac{1}{\pi} \int_{-\sqrt{2}}^{\sqrt{2}} \log(x-y)\sqrt{2-x^2} dx = \frac{x^2}{2} - \frac{x}{2}\sqrt{x^2-2} + \log\left(\frac{x+\sqrt{x^2-2}}{2}\right) - \frac{1}{2}.$$

By an easy change of variables, we get that

$$F(x) = L\left(\frac{x}{\sqrt{2}}\right) + \frac{1}{2}\log 2,$$

so that

$$\frac{x^2}{2} - 2F(x) = \frac{x}{2}\sqrt{x^2-4} - 2\log\left(\frac{x+\sqrt{x^2-4}}{2}\right) + 1 = \int_2^x \sqrt{y^2-4} dy + 1.$$

By continuity, we get that the constant in the previous computation was $C = 1$ and that both parts of the lemma hold.

## C  On strict convexity of the logarithmic energy

The object of this appendix is to prove Lemma 1.9. As for the definition of $I$ in (5), we restrict ourselves to probability measures $\mu$ such that $\int x^2 d\mu(x) < \infty$.

The idea is that the rate function $I$ is the difference of a linear term $\mu \mapsto \int x^2 d\mu(x)$ and a functional $\Sigma : \mu \mapsto \int\int \log|x-y| d\mu(x) d\mu(y)$, which is essentially strictly concave. Following the proof of Lemma 2.6.2. in [3], we use a slightly different decomposition of $I$.

By using the fact that $\mu_{\text{sc}}$ satisfies the EL equations, one can rewrite

$$I(\mu) = -\Sigma(\mu - \mu_{\text{sc}}) + \int \left( \frac{x^2}{2} - 2 \int \log|x - y| \, \mathrm{d}\mu_{\text{sc}}(y) - 1 \right) \mathrm{d}\mu(x).$$

The second term is linear in $\mu$ and we will now prove the strict concavity of $\mu \mapsto \Sigma(\mu - \mu_{\text{sc}})$.

We choose an appropriate representation of the logarithm: from the equality

$$\frac{1}{z} = \frac{1}{2z} \int_0^\infty e^{-\frac{u}{2}} \mathrm{d}u,$$

which holds for any $z \in \mathbb{R}^*$ and using the change of variables $u = \frac{z^2}{t}$, we get

$$\frac{1}{z} = \frac{z}{2} \int_0^\infty e^{-\frac{z^2}{2t}} \frac{\mathrm{d}t}{t^2}.$$

For $x \neq y$, integrating from 1 to $|x - y|$, we get

$$\log|x - y| = \int_1^{|x-y|} \frac{z}{t} \int_0^\infty e^{-\frac{z^2}{2t}} \frac{\mathrm{d}t}{t} \mathrm{d}z = \int_0^\infty \frac{e^{-\frac{1}{2t}} - e^{-\frac{|x-y|^2}{2t}}}{2t} \mathrm{d}t.$$

As $\mu - \mu_{\text{sc}}$ has mass zero, the first term will cancel and we get the following Fourier representation

$$\Sigma(\mu - \mu_{\text{sc}}) = -\int_0^\infty \frac{1}{2t} \left( \iint e^{-\frac{|x-y|^2}{2t}} \mathrm{d}(\mu - \mu_{\text{sc}})(x) \mathrm{d}(\mu - \mu_{\text{sc}})(y) \right) \mathrm{d}t$$

$$= -\int_0^\infty \sqrt{\frac{t}{2\pi}} \int_{-\infty}^\infty \left| \int e^{i\lambda x} \mathrm{d}(\mu - \mu_{\text{sc}})(x) \right|^2 e^{-\frac{t\lambda^2}{2}} \mathrm{d}\lambda.$$

Now $\mu \mapsto \left| \int e^{i\lambda x} \mathrm{d}(\mu - \mu_{\text{sc}})(x) \right|^2$ is convex so that $\mu \mapsto \Sigma(\mu - \mu_{\text{sc}})$ is concave.

Moreover, for $\alpha \in [0, 1]$ and any probability measures $\mu$ and $\nu$ so that $\Sigma$ is well defined, we have

$$\Sigma(\alpha\mu + (1 - \alpha)\nu) = \alpha\Sigma(\mu) + (1 - \alpha)\Sigma(\nu) + (\alpha^2 - \alpha)\Sigma(\mu - \nu).$$

From the Fourier representation above, we know that $\Sigma(\mu - \nu) \geq 0$ and $\Sigma(\mu - \nu) = 0$ if and only if all Fourier coefficients are zero, that is if $\mu = \nu$.

This concludes the proof of the strict convexity.

**Acknowledgements** We warmly acknowledge the organizers of Les Houches 2024 summer school, and especially Grégory Schehr. We also thank Raphaël Ducatez for sharing part of his notes taken during the lectures and for fruitful discussions.

**Funding information** JLF acknowledges ANR project ESQuisses, grant number ANR-20-CE47-0014-01 and ANR project Quantum Trajectories, grant number ANR-20-CE40-0024-01. MM acknowledges the support of Labex CEMPI, grant number ANR-1-LABX-0007-01, of the RT Mathématiques et Physique, funded by CNRS Mathématiques and by CNRS Physique and of the CDP C2EMPI, together with the French State under the France-2030 programme, the University of Lille, the Initiative of Excellence of the University of Lille, the European Metropolis of Lille for their funding and support of the R-CDP-24-004-C2EMPI project.

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
