# Peer review of "Large deviations in Coulomb gases: a mathematical perspective"

_SciPost Physics Lecture Notes_

## Round 1 · Referee Report · Anonymous (Referee 1) · 2025-5-26

Strengths

Highest quality lecture material written by benevolent mathematicians for theoretical physicists. It exposes the way probabilists think about
large deviation problems familiar to physicists in a way that allows to learn the corresponding notations and grasp the essence of methods of probabilistic reasoning.

Weaknesses

There are essentially none, some very minor suggestions

Report

These high-quality lecture notes provide a detailed overview of the topics of large deviations in Coulomb gases, with emphasis on random matrix appllications. They are written by mathematicians and aimed at theoretical physicists interested in learning the methodology of large deviations on examples which are in general well-understood by physisicsts, but in a language which is rather different from their own.
The authors should be highly praised for their efforts invested in writing a meticulous exposition of the essential mathematical tools and presenting proofs in a friendly way which indeed could be understood by devoted reader with theoretical physics background, if she or he is prepared to invest their time in following the exposition. I have no doubts the notes will be a precious addition facilitating the ongoing fruitful dialogue between the two communities. I recommend publication of the notes in the strongest terms.

Requested changes

A few points which perhaps could be considered for change (optionally): 1. Proposition 1.3, eq.(2): why in the beginning we denote eigenvalues $\lambda_i$, but then use $x_i$ in Eq.(2) instead of $\lambda_i$? It is a bit confusing. Perhaps one should use x_i everywhere?

2) Discussion of the notion of tightness in page 9 is nice, but to appreciate importance of this notion and to develop an intuition further it would be very helpful to provide a simple illustrative example of a sequence of measures which is not tight.

3) Perhaps physicists might need a reminder of the notation [...]^c being complement of the set [...] and the meaning of x^2 ∧ Mdμ

4) When formulating Lemma 1.7 I would suggest: Any minimizer μ of the rate funtion I [ AS DEFINED in EQ.(5)]

5) line 642: H_\mu - what is the name for it? Is it not what is known as the Hilbert transform? If it is, the name would be useful to mention.

6) In page 30 is nice to see how mathematicians approach the heuristics which is frequently used in the physics literature without much ado: the vector uniformly distributed over the high-dimensional unit sphere is statistically equivalent to a gaussian-distributed vector. Perhaps this fact deserves to be mentioned.

Recommendation

Publish (surpasses expectations and criteria for this Journal; among top 10%)

---

## Round 1 · Referee Report · Anonymous (Referee 2) · 2025-10-30

Disclosure of Generative AI use

The referee discloses that the following generative AI tools have been used in the preparation of this report:

ChatGPT found the very minor typos (ChatGPT5, free version, used on October 30, 2025)

Strengths

This is a very pedagogical introduction to large deviations of the eigenvalue distribution of random matrices with a focus on mathematical methods and analytical subtleties. Authors start with the standard case of the GUE, which suggests a more general Coulomb gas approach for the large deviations of the full eigenvalue empirical distribution, and then they discuss a method based on spherical integrals to study large deviations of the largest eigenvalue. There is the right balance of details and intuition.

Weaknesses

none

Report

I believe that these lecture notes are an excellent contribution to the journal. I only list a few comments below that the authors might take into account.

1- It might be counterintuitive that the notation $\mu_{GUE_N}$ for the probability distribution on matrices is so different from the notation $\mathbb P_{GUE_N}$ for the marginal probability distribution of the eigenvalues. It may cause some confusion with $\hat{\mu}_N$ which is really different (a random probability measure). Even though mathematician probably prefer to avoid notation overload, I would suggest to use the same notation $\mathbb P_{GUE_N}$ for both the GUE probability measure on matrices and its eigenvalue density. I would also suggest to replace the $x_i$ in the GUE density formula (Proposition 1.3) by $\lambda_i$, so as to avoid confusion with the real part of the entries discussed a few lines above.

2- Below theorem 3.3 the subsection title is "Unitarily invariant deformed random matrices" but then the matrix O is orthogonal. Moreover footnote 15 on page 33 is about the Haar measure on the unitary group. In this section, it is not always perfectly clear if we work with the unitary group, or orthogonal group, or if it does not matter. It seems that modulo trivial details, the only place where one really needs the unitary group is for the HCIZ formula, but this could be clarified.

Requested changes

Other very minor typos:

1- page 5: correspondance -> correspondence 2- page 6 phycisists 3- page 6 adressed -> addressed 4- page 11 satifies -> satisfies 5- page 20 Equation (11) double parenthesis around x_1, ..., x_N 6-please choose between the spellings "minimizer" and "minimiser", for consistency.

Recommendation

Publish (meets expectations and criteria for this Journal)

---

## Editorial Decision

refereeing_in_preparation